# Visual Correspondence Hallucination

**Hugo Germain[1], Vincent Lepetit[1] and Guillaume Bourmaud[2]**
[1]LIGM, École des Ponts, Univ Gustave Eiffel, CNRS, Marne-la-Vallée, France
[2]IMS, University of Bordeaux, Bordeaux INP, CNRS, Bordeaux, France
`{firstname.lastname}@enpc.fr, guillaume.bourmaud@u-bordeaux.fr`

## Abstract

Given a pair of partially overlapping source and target images and a keypoint in the source image, the keypoint's correspondent in the target image can be either visible, occluded or outside the field of view. Local feature matching methods are only able to identify the correspondent's location when it is visible, while humans can also hallucinate (*i.e.* predict) its location when it is occluded or outside the field of view through geometric reasoning. In this paper, we bridge this gap by training a network to output a peaked probability distribution over the correspondent's location, regardless of this correspondent being visible, occluded, or outside the field of view. We experimentally demonstrate that this network is indeed able to hallucinate correspondences on pairs of images captured in scenes that were not seen at training-time. We also apply this network to an absolute camera pose estimation problem and find it is significantly more robust than state-of-the-art local feature matching-based competitors.

## 1 Introduction

Establishing correspondences between two partially overlapping images is a fundamental computer vision problem with many applications. For example, state-of-the-art methods for visual localization from an input image rely on keypoint matches between the input image and a reference image (Sattler et al., 2018; Sarlin et al., 2019; 2020; Revaud et al., 2019). However, these local feature matching methods will still fail when few keypoints are *covisible*, *i.e.* when many image locations in one image are outside the field of view or become occluded in the second image. These failures are to be expected since these methods are pure pattern recognition approaches that seek to *identify* correspondences, *i.e.* to find correspondences in covisible regions, and consider the non-covisible regions as noise. By contrast, humans explain the presence of these non-covisible regions through geometric reasoning and consequently are able to *hallucinate* (*i.e.* predict) correspondences at those locations. Geometric reasoning has already been used in computer vision for image matching, but usually as an *a posteriori* processing (Fischler & Bolles, 1981; Luong & Faugeras, 1996; Barath & Matas, 2018; Chum et al., 2003; 2005; Barath et al., 2019; 2020). These methods seek to remove outliers from the set of correspondences produced by a local feature matching approach using only limited geometric models such as epipolar geometry or planar assumptions.

**Contributions.** In this paper we tackle the problem of correspondence hallucination. In doing so we seek to answer two questions: $(i)$ can we derive a network architecture able to learn to hallucinate correspondences? and $(ii)$ is correspondence hallucination beneficial for absolute pose estimation? The answer to these questions is the main novelty of this paper. More precisely, we consider a network that takes as input a pair of partially overlapping source/target images and keypoints in the source image, and outputs for each keypoint a probability distribution over its correspondent's location in the target image plane. We propose to train this network to both identify and hallucinate the keypoints' correspondents. We call the resulting method NeurHal, for Neural Hallucinations. To the best of our knowledge, learning to hallucinate correspondences is a virgin territory, thus we first provide an analysis of the specific features of that novel learning task. This analysis guides us towards employing an appropriate loss function and designing the architecture of the network. After training the network, we experimentally demonstrate that it is indeed able to hallucinate correspondences on unseen pairs of images captured in novel scenes. We also apply this network to a camera pose estimation problem and find it is significantly more robust than state-of-the-art local feature matching-based competitors.

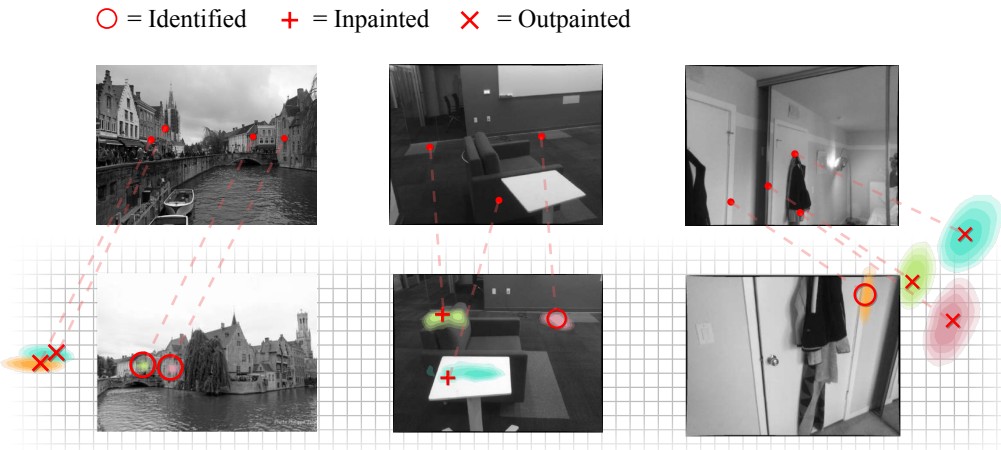

Figure 1: **Visual correspondence hallucination.** Our network, called NeurHal, takes as input a pair of partially overlapping source and target images and a set of keypoints detected in the source image, and outputs for each keypoint a probability distribution over its correspondent's location in the target image. When the correspondent is actually visible, its location can be *identified*; when it is not, its location must be *hallucinated*. Two types of hallucination tasks can be distinguished: 1) if the correspondent is occluded, its location has to be *inpainted*; 2) if it is outside the field of view of the target image, its location needs to be *outpainted*. NeurHal generalizes to scenes not seen during training: For each of these three pairs of source/target images coming from the test scenes of ScanNet (Dai et al., 2017) and MegaDepth (Li & Snavely, 2018), we show (top row) the source image with a small subset of keypoints, and (bottom row) the target image with the probability distributions predicted by our network and the ground truth correspondents: o for the identified correspondents, +️ for the inpainted ones, and ✕ for the outpainted correspondents.

## 2  RELATED WORK

To the best of our knowledge, aiming at hallucinating visual correspondences has never been done but the related fields of local feature description and matching are immensely vast, and we focus here only on recent learning-based approaches.

**Learning-based local feature description.** Using deep neural networks to learn to compute local feature descriptors have shown to bring significant improvements in invariance to viewpoint and illumination changes compared to handcrafted methods (Csurka & Humenberger, 2018; Gauglitz et al., 2011; Salahat & Qasaimeh, 2017; Balntas et al., 2017). Most methods learn descriptors locally around pre-computed *covisible* interest regions in both images (Yi et al., 2016; Detone et al., 2018; Balntas et al., 2016a; Luo et al., 2019), using convolutional-based siamese architectures trained with a contrastive loss (Gordo et al., 2016; Schroff et al., 2015; Balntas et al., 2016b; Radenović et al., 2016; Mishchuk et al., 2017; Simonyan et al., 2014), or using pose (Wang et al., 2020; Zhou et al., 2021) or self (Yang et al., 2021) supervision. To further improve the performances, (Dusmanu et al., 2019; Revaud et al., 2019) propose to jointly learn to detect and describe keypoints in both images, while Germain et al. (2020) only detects in one image and densely matches descriptors in the other.

**Learning-based local feature matching.** All the methods described in the previous paragraph establish correspondences by comparing descriptors using a simple operation such as a dot product. Thus the combination of such a simple matching method with a siamese architecture inevitably produces outlier correspondences, especially in non-covisible regions. To reduce the amount of outliers, most approaches employ so-called Mutual Nearest Neighbor (MNN) filtering. However, it is possible to go beyond a simple MNN and learn to match descriptors. Learning-based matching methods (Zhang et al., 2019; Brachmann & Rother, 2019; Moo Yi et al., 2018; Sun et al., 2020; Choy et al., 2020; 2016) take as input local descriptors and/or putative correspondences, and learn to output correspondences probabilities. However, all these matching methods focus only on predicting correctly covisible correspondences.

**Jointly learning local feature description and matching.** Several methods have recently proposed to jointly learn to compute and match descriptors (Sarlin et al., 2020; Sun et al., 2021; Li et al., 2020; Rocco et al., 2018; 2020). All these methods use a siamese Convolutional Neural Network (CNN) to obtain dense local descriptors, but they significantly differ regarding the way they establish matches. They actually fall into two categories. The first category of methods (Li et al., 2020; Rocco et al., 2018; 2020) computes a 4D correlation tensor that essentially represents the scores of all the possible correspondences. This 4D correlation tensor is then used as input to a second network that learns to modify it using soft-MNN and 4D convolutions. Instead of summarizing all the information into a 4D correlation tensor, the second category of methods (Sarlin et al., 2020; Sun et al., 2021) rely on Transformers (Vaswani et al., 2017; Dosovitskiy et al., 2020; Ramachandran et al., 2019; Caron et al., 2021; Cordonnier et al., 2020; Zhao et al., 2020; Katharopoulos et al., 2020) to let the descriptors of both images communicate and adapt to each other. All these methods again focus on identifying correctly covisible correspondences and consider non-covisible correspondences as noise. While our architecture is closely related to the second category of methods as we also rely on Transformers, the motivation for using it is quite different since it is our goal of hallucinating correspondences that calls for a non-siamese architecture (see Sec.3).

**Visual content hallucination.** (Yang et al., 2019) proposes to hallucinate the content of RGB-D scans to perform relative pose estimation between two images. More recently (Chen et al., 2021) regresses distributions over relative camera poses for spherical images using joint processing of both images. The work of (Yang et al., 2020; Qian et al., 2020; Jin et al., 2021) shows that employing a *hallucinate-then-match* paradigm can be a reliable way of recovering 3D geometry or relative pose from sparsely sampled images. In this work, we focus on the problem of *correspondence* hallucination which unlike previously mentioned approaches does not aim at recovering explicit visual content or directly regressing a camera pose. Perhaps closest to our goal is Cai et al. (2021) that seeks to estimate a relative rotation between two non-overlapping images by learning to reason about "hidden" cues such as direction of shadows in outdoor scenes, parallel lines or vanishing points.

## 3 OUR APPROACH

Our goal is to train a network that takes as input a pair of partially overlapping source/target images and keypoints in the source image, and outputs for each keypoint a probability distribution over its correspondent's location in the target image plane, regardless of this correspondent being visible, occluded, or outside the field of view. While the problem of learning to find the location of a *visible* correspondent received a lot of attention in the past few years (see Sec. 2), to the best of our knowledge, this paper is the first attempt of learning to find the location of a correspondent regardless of this correspondent being visible, occluded, or outside the field of view. Since this learning task is virgin territory, we first analyze its specific features below, before defining a loss function and a network architecture able to handle these features.

### 3.1 ANALYSIS OF THE PROBLEM

The task of finding the location of a correspondent regardless of this correspondent being visible, occluded, or outside the field of view actually leads to three different problems. Before stating those three problems, let us first recall the notion of correspondent as it is the keystone of our problem.

**Correspondent.** Given a keypoint $\mathbf{p}_\mathrm{S} \in \mathbb{R}^2$ in the source image $\mathtt{I}_\mathrm{S}$, its depth $d_\mathrm{S} \in \mathbb{R}^+$, and the relative camera pose $\mathtt{R}_\mathrm{TS} \in \mathrm{SO}(3)$, $\mathbf{t}_\mathrm{TS} \in \mathbb{R}^3$ between the coordinate systems of $\mathtt{I}_\mathrm{S}$ and the target image $\mathtt{I}_\mathrm{T}$, the *correspondent* $\mathbf{p}_\mathrm{T} \in \mathbb{R}^2$ of $\mathbf{p}_\mathrm{S}$ in the target image plane is obtained by warping $\mathbf{p}_\mathrm{S}$: $\mathbf{p}_\mathrm{T} := \omega\left(d_\mathrm{S}, \mathbf{p}_\mathrm{S}, \mathtt{R}_\mathrm{TS}, \mathbf{t}_\mathrm{TS}\right) := \mathtt{K}_\mathrm{T}\pi\left(d_\mathrm{S}\mathtt{R}_\mathrm{TS}\mathtt{K}_\mathrm{S}^{-1}\mathbf{p}_\mathrm{S} + \mathbf{t}_\mathrm{TS}\right)$, where $\mathtt{K}_\mathrm{S}$ and $\mathtt{K}_\mathrm{T}$ are the camera calibration matrices of source and target images and $\pi\left(\mathbf{u}\right) := \left[\mathbf{u}_x/\mathbf{u}_z, \mathbf{u}_y/\mathbf{u}_z, 1\right]^\mathsf{T}$ is the projection function. In a slight abuse of notation, we do not distinguish a homogeneous 2D vector from a non-homogeneous 2D vector. Let us highlight that the correspondent $\mathbf{p}_\mathrm{T}$ of $\mathbf{p}_\mathrm{S}$ may not be *visible*, *i.e.* it may be occluded or outside the field of view.

**Identifying the correspondent.** In the case where a network has to establish a correspondence between a keypoint $\mathbf{p}_\mathrm{S}$ in $\mathtt{I}_\mathrm{S}$ and its *visible* correspondent $\mathbf{p}_\mathrm{T}$ in $\mathtt{I}_\mathrm{T}$, standard approaches, such as

comparing a local descriptor computed at $\mathbf{p}_S$ in $\mathtt{I}_\mathtt{S}$ with local descriptors computed at detected keypoints in $\mathtt{I}_\mathtt{T}$, are applicable to *identify* the correspondent $\mathbf{p}_\mathtt{T}$.

**Outpainting the correspondent.** When $\mathbf{p}_\mathtt{T}$ is outside the field of view of $\mathtt{I}_\mathtt{T}$, there is nothing to identify, *i.e.* neither can $\mathbf{p}_\mathtt{T}$ be detected as a keypoint nor can a local descriptor be computed at that location. Here the network first needs to identify correspondences in the region where $\mathtt{I}_\mathtt{T}$ overlaps with $\mathtt{I}_\mathtt{S}$ and realize that the correspondent $\mathbf{p}_\mathtt{T}$ is outside the field of view to eventually *outpaint* it (see Fig. 1). We call this operation "outpainting the correspondent" as the network needs to predict the location of $\mathbf{p}_\mathtt{T}$ outside the field of view of $\mathtt{I}_\mathtt{T}$.

**Inpainting the correspondent.** When $\mathbf{p}_\mathtt{T}$ is occluded in $\mathtt{I}_\mathtt{T}$, the problem is even more difficult since local features can be computed at that location but will not match the local descriptor computed at $\mathbf{p}_\mathtt{S}$ in $\mathtt{I}_\mathtt{S}$. As in the outpainting case, the network needs to identify correspondences in the region where $\mathtt{I}_\mathtt{T}$ overlaps with $\mathtt{I}_\mathtt{S}$ and realize that the correspondent $\mathbf{p}_\mathtt{T}$ is occluded to eventually *inpaint* the correspondent $\mathbf{p}_\mathtt{T}$ (see Fig. 1). We call this operation "inpainting the correspondent" as the network needs to predict the location of $\mathbf{p}_\mathtt{T}$ behind the occluding object.

Let us now introduce a loss function and an architecture that are able to unify the identifying, inpainting and outpainting tasks.

### 3.2 Loss function

The distinction we made between the identifying, inpainting and outpainting tasks come from the fact that the source image $\mathtt{I}_\mathtt{S}$ and the target image $\mathtt{I}_\mathtt{T}$ are the projections of the same 3D environment from two different camera poses. In order to integrate this idea and obtain a unified correspondence learning task, we rely on the *Neural Reprojection Error* (NRE) introduced by (Germain et al., 2021). In order to properly present the NRE, we first recall the notion of *correspondence map*.

**Correspondence map.** Given $\mathtt{I}_\mathtt{S}$, $\mathtt{I}_\mathtt{T}$ and a keypoint $\mathbf{p}_\mathtt{S}$ in the image plane of $\mathtt{I}_\mathtt{S}$, the *correspondence map* $\mathtt{C}_\mathtt{T}$ of $\mathbf{p}_\mathtt{S}$ in the image plane of $\mathtt{I}_\mathtt{T}$ is a 2D tensor of size $H_\mathtt{C} \times W_\mathtt{C}$ such that $\mathtt{C}_\mathtt{T}(\mathbf{p}_\mathtt{T}) := p(\mathbf{p}_\mathtt{T}|\mathbf{p}_\mathtt{S}, \mathtt{I}_\mathtt{S}, \mathtt{I}_\mathtt{T})$ is the likelihood of $\mathbf{p}_\mathtt{T}$ being the correspondent of $\mathbf{p}_\mathtt{S}$. The likelihood can only be evaluated for $\mathbf{p}_\mathtt{T} \in \Omega_{\mathtt{C}_\mathtt{T}}$ where $\Omega_{\mathtt{C}_\mathtt{T}}$ is the set of all the pixel locations in $\mathtt{C}_\mathtt{T}$. Here, we implicitly defined that the likelihood of $\mathbf{p}_\mathtt{T}$ falling outside the boundaries of $\mathtt{C}_\mathtt{T}$ is zero. In practice, a correspondence map $\mathtt{C}_\mathtt{T}$ is implemented as a neural network that takes as input $\mathbf{p}_\mathtt{S}$, $\mathtt{I}_\mathtt{S}$ and $\mathtt{I}_\mathtt{T}$, and outputs a softmaxed 2D tensor. A correspondence map $\mathtt{C}_\mathtt{T}$ may not have the same number of lines and columns than $\mathtt{I}_\mathtt{T}$ especially when the goal is to outpaint a correspondence. Thus, in the general case, to transform a 2D point from the image plane of $\mathtt{I}_\mathtt{T}$ to the correspondence plane of $\mathtt{C}_\mathtt{T}$, we will need another affine transformation matrix $\mathtt{K}_\mathtt{C}$. Let us highlight that this likelihood is obtained using the visual content of $\mathtt{I}_\mathtt{S}$ and $\mathtt{I}_\mathtt{T}$ only.

**Neural Reprojection Error.** The NRE (Germain et al., 2021) is a loss function that warps a keypoint $\mathbf{p}_\mathtt{S}$ into the image plane of $\mathtt{I}_\mathtt{T}$ and evaluates the negative log-likelihood at this location. In our context, the NRE can be written as:

$$\mathrm{NRE}\left(\mathbf{p}_\mathtt{S}, \mathtt{C}_\mathtt{T}, \mathtt{R}_\mathtt{TS}, \mathbf{t}_\mathtt{TS}, d_\mathtt{S}\right) := -\ln \mathtt{C}_\mathtt{T}\left(\mathbf{x}_\mathtt{T}\right) \text{ where } \mathbf{x}_\mathtt{T} = \mathtt{K}_\mathtt{C}\,\omega\left(d_\mathtt{S}, \mathbf{p}_\mathtt{S}, \mathtt{R}_\mathtt{TS}, \mathbf{t}_\mathtt{TS}\right) . \quad (1)$$

In general, $\mathbf{x}_\mathtt{T}$ does not have integer coordinates and the notation $\ln \mathtt{C}_\mathtt{T}(\mathbf{x}_\mathtt{T})$ corresponds to performing a bilinear interpolation *after* the logarithm. For more details concerning the derivation of the NRE, the reader is referred to Germain et al. (2021).

The NRE provides us with a framework to learn to identify, inpaint or outpaint the correspondent of $\mathbf{p}_\mathtt{S}$ in $\mathtt{I}_\mathtt{T}$ in a unified manner since Eq. (1) is differentiable *w.r.t.* $\mathtt{C}_\mathtt{T}$ and there is no assumption regarding covisibility. The main difficulty to overcome is the definition of a network architecture able to output a consistent $\mathtt{C}_\mathtt{T}$ being given only $\mathbf{p}_\mathtt{S}$, $\mathtt{I}_\mathtt{S}$ and $\mathtt{I}_\mathtt{T}$ as inputs, *i.e.* the network must figure out whether the correspondent of $\mathbf{p}_\mathtt{S}$ in $\mathtt{I}_\mathtt{T}$ can be identified or has to be inpainted or outpainted.

### 3.3 Network architecture

The analysis from Sec. 3.1 and the use of the NRE as a loss (Sec. 3.2) call for:
• a non-siamese architecture to be able to link the information from $\mathtt{I}_\mathtt{S}$ with the information from $\mathtt{I}_\mathtt{T}$

to *outpaint* or *inpaint* the correspondent if needed;
• an architecture that outputs a matching score for all the possible locations in $I_T$ as well as locations beyond the field of view of $I_T$ as the network could decide to identify, inpaint or outpaint a correspondent at these locations.

To fulfill these requirements, we propose the following: Our network takes as input $I_S$ and $I_T$ as well as a set of keypoints $\{\mathbf{p}_{S,n}\}_{n=1...N}$ in the source image plane of $I_S$. A siamese CNN backbone is applied to $I_S$ and $I_T$ to produce compact dense local descriptor maps $H_S$ and $H_T$. In order to be able to *outpaint* correspondents in the target image plane, we pad $H_T$ with a learnable fixed vector $\boldsymbol{\lambda}$. This padding step allows to *initialize* descriptors at locations outside the field of view of $I_T$. We note $\gamma$ the relative output-to-input correspondence map resolution ratio.

The dense descriptor maps $H_S$ and $H_{T,pad}$, and the keypoints $\{\mathbf{p}_{S,n}\}_{n=1...N}$ are then used as inputs of a cross-attention-based backbone $\mathcal{F}$ with positional encoding. This part of the network outputs a feature vector $\mathbf{d}_{S,n}$ for each keypoint $\mathbf{p}_{S,n}$ and dense feature vectors $D_{T,pad}$ of the size of $H_{T,pad}$. This cross-attention-based backbone allows the local descriptors $H_S$ and $H_{T,pad}$ to *communicate* with each other. Thus, during training, the network will be able to leverage this ability to communicate, to learn to *hallucinate* peaked *inpainted* and *outpainted* correspondence maps.

The correspondence map $C_{T,n}$ of $\mathbf{p}_{S,n}$ in the image plane of $I_T$ is computed by applying a $1\times1$ convolution to $D_{T,pad}$ using $\mathbf{d}_{S,n}$ as filter, followed by a 2D softmax.

An overview of our architecture, that we call NeurHal, is presented in Fig. 2. In practice, in order to keep the required amount of memory and the computational time reasonably low, the correspondence maps $\{C_{T,n}\}_{n=1...N}$ have a low resolution, *i.e.* for a target image of size $640 \times$

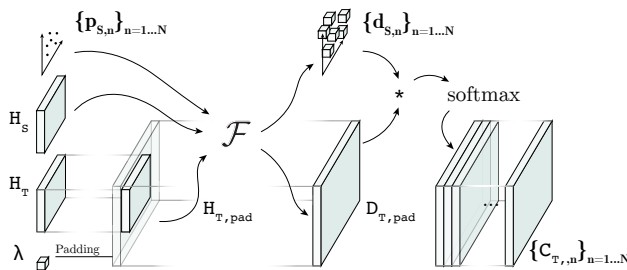

Figure 2: **Overview of NeurHal:** See text for details.

$480$, we use a CNN with an effective stride of $s = 8$ and consequently the resulting correspondence maps (with $\gamma = 50\%$) are of size $160 \times 120$. Producing low resolution correspondence maps prevents NeurHal from predicting accurate correspondences. But as we show in the experiments, this low resolution is sufficient to hallucinate correspondences and have an *affirmative answer* to both questions: (i) can we derive a network architecture able to learn to hallucinate correspondences? and (ii) is correspondence hallucination beneficial for absolute pose estimation? Thus, we leave the question of the accuracy of hallucinated correspondences for future research. Additional details concerning the architecture are provided in Sec. C.1 of the appendix.

## 3.4 TRAINING-TIME

Given a pair of partially overlapping images $(I_S, I_T)$, a set of keypoints with ground truth depths $\{\mathbf{p}_{S,n}, d_{S,n}\}_{n=1...N}$ as well as the ground truth relative camera pose $(R_{TS}, \mathbf{t}_{TS})$, the corresponding sum of NRE terms (Eq. 1) can be minimized *w.r.t.* the parameters of the network that produces the correspondence maps. Thus, we train our network using stochastic gradient descent and early stopping by providing pairs of overlapping images along with the aforementioned ground truth information. Let us also highlight that there is no distinction in the training process between the identifying, inpainting and outpainting tasks since the only thing our network outputs are correspondence maps. Moreover there is no need for labeling keypoints with ground truth labels such as "identify/visible", "inpaint/occluded" or "outpaint/outside the field of view". Additional information concerning the training are provided in Sec. C.2 of the appendix.

## 3.5 TEST-TIME

At test-time, our network only requires a pair of partially overlapping images $(I_S, I_T)$ as well as keypoints $\{\mathbf{p}_{S,n}\}_{n=1...N}$ in $I_S$, and outputs a correspondence map $C_{T,n}$ in the image plane of $I_T$ for each keypoint, regardless of its correspondent being visible, occluded or outside the field of view.

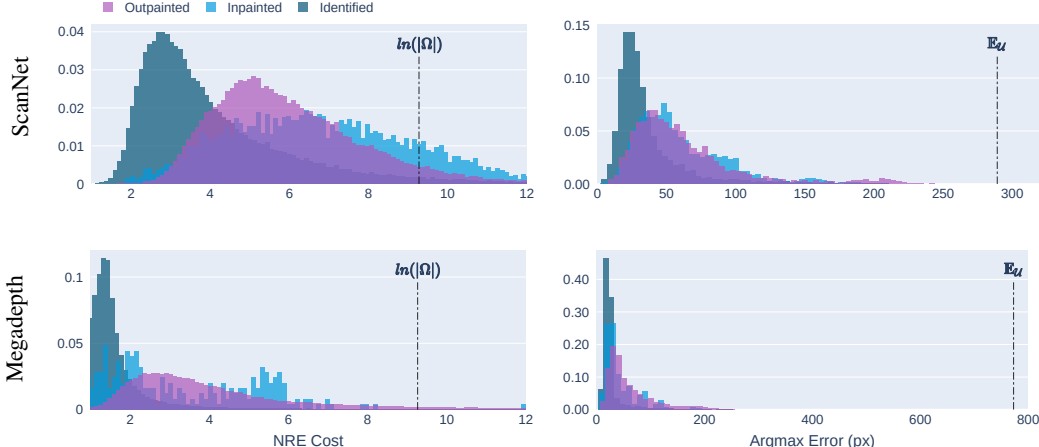

Figure 3: **Evaluation of the ability of NeurHal to hallucinate correspondences on the test scenes of ScanNet and MegaDepth.** (left) Histograms of the NRE (see Eq. 1) for each task (identifying, outpainting, inpainting), computed on correspondence maps produced by NeurHal. The value $\ln |\Omega_{c_\tau}|$ is the NRE of a uniform correspondence map. (right) Histograms of the errors between the argmax (mode) of a correspondence map and the ground truth correspondent's location, for each task. The value $\mathbb{E}_{\mathcal{U}}$ is the average error of a random prediction.

## 4 EXPERIMENTS

In these experiments, we seek to answer two questions: 1) "Is the proposed NeurHal approach presented in Sec. 3 capable of hallucinating correspondences?" and 2) "In the context of absolute camera pose estimation, does the ability to hallucinate correspondences bring further robustness?".

### 4.1 EVALUATION OF THE ABILITY TO HALLUCINATE CORRESPONDENCES

We evaluate the ability of our network to hallucinate correspondences on four datasets: the indoor datasets ScanNet (Dai et al., 2017) and NYU (Nathan Silberman & Fergus, 2012), and the outdoor datasets MegaDepth (Li & Snavely, 2018) and ETH-3D (Schöps et al., 2017). For the indoor setting (outdoor setting, respectively), we train NeurHal on ScanNet (Megadepth, respectively) on the training scenes as described in Sec. 3.4, and evaluate it on the *disjoint* set of validation scenes. Thus, all the qualitative and quantitative results presented in this section cannot be ascribed to scene memorization. For each dataset, we run predictions over 2, 500 source and target image pairs sampled from the test set, with overlaps between 2% and 80%. For every image pair, we also feed as input to NeurHal keypoints in the source image. These keypoints have known ground truth correspondents in the target image and labels (visible, occluded, outside the field of view) that we use to evaluate the ability of our network to hallucinate correspondences. For more details on the settings of our experiment see Sec. C.2. For this experiment, we use $\gamma = 50\%$.

We report in Fig. 3 two histograms computed over more than one million keypoints for each task we seek to validate: identification, inpainting, and outpainting. The first histogram Fig. 3 (left) is obtained by evaluating for each correspondence map the NRE cost (Eq. 1) at the ground truth correspondent's location. In order to draw conclusions, we also report the negative log-likelihood of a uniform correspondence map ($\ln |\Omega_{c_\tau}|$). We find that for each task and for both datasets, the predicted probability mass lies significantly below $\ln |\Omega_{c_\tau}|$, which demonstrates NeurHal's ability to perform identification, inpainting and outpainting. On ScanNet, we also observe that identification is a simpler task than outpainting while inpainting is the hardest task: On average, the NRE cost of inpainted correspondents is higher than the average NRE cost of outpainted correspondents, which indicates the predicted correspondence maps are less peaked for inpainting than they are for outpainting. This corroborates what we empirically observed on qualitative results in Fig. 1, and supports our analysis in Sec. 3.1. On Megadepth, outpainting and inpainting histograms have a similar shape which does

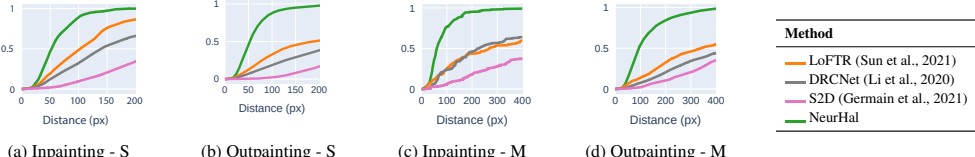

(a) Inpainting - S  (b) Outpainting - S  (c) Inpainting - M  (d) Outpainting - M

Figure 4: **Ability to hallucinate - comparison against state-of-the-art local feature matching methods on ScanNet (S) and Megadepth (M).** For each method, we report the percentage of keypoint's correspondents whose distance *w.r.t.* the ground truth location is lower than $x$ pixels, as a function of $x$, for (a-c) the inpainting task and (b-d) the outpainting task.

not reflect the previous statement, but we believe this is due to the fact that inpainting labels are noisy for this dataset, as explained in Sec. C.2.

On the right histogram of Fig. 3, we report the distribution of the distance between the argmax of a correspondence map and the ground truth correspondent's location. We also report the average error of a random prediction. We find the histogram mass lies significantly to the left of the random prediction average error, indicating our model is able to place modes correctly in the correspondence maps, regardless of the task at hand. On ScanNet, we observe that the inpainting and outpainting histograms are very similar, indicating the predicted argmax is equally good for both tasks. As mentioned above, the correspondence maps produced by NeurHal have a low resolution (see Sec. 3.3) which explains why the "argmax error" is not closer to zero pixel.

In Fig. 4, we compare the hallucination performances of NeurHal against state-of-the-art local feature matching methods. Since all these local feature matching methods were designed and trained on pairs of images with significant overlap to perform only identification, they obtain poor inpainting results. Concerning the outpainting task, these methods seek to find a correspondent within the image boundaries, consequently they cannot outpaint correspondences and obtain very poor results.

In Fig. 5 we show several qualitative inpainting/outpainting results on ScanNet and MegaDepth datasets. In the appendix, we also report qualitative results obtained on the NYU Depth dataset (Fig. 16) and on the ETH-3D dataset (Fig. 15).

These results allow us to conclude that NeurHal is able to hallucinate correspondences with a strong generalization capacity. Additional experiments concerning the ability to hallucinate correspondences are provided in Sec. A as well as technical details regarding the evaluation protocol in Sec. C.3.

## 4.2 APPLICATION TO ABSOLUTE CAMERA POSE ESTIMATION

In the previous experiment, we showed that our network is able to hallucinate correspondences. We now evaluate whether this ability helps improving the robustness of an absolute camera pose estimator. We run this evaluation on the test set of ScanNet over 2,500 source and target image pairs captured in scenes that were not used at training time. For each source/target image pair, we employ NeurHal to produce correspondence maps. As in the previous experiment, we use $\gamma = 50\%$. Given these correspondence maps and the depth map of the source image, we estimate the absolute camera pose between the target image and the source image using the method proposed in Germain et al. (2021).

In Fig. 6, we show the results of an ablation study conducted on ScanNet. In this study, we focus on the robustness of the camera pose estimate for various combinations of training data, *i.e.* we consider a pose is "correct" if the rotation error is lower than 20 degrees and the translation error is below 1.5 meters (see Sec. C.3). We find that training our network to perform the three tasks (identification, inpainting, and outpainting) produces the best results. In particular, we find that adding outpainting plays a critical role in improving localization of low-overlap image pairs. We also find that learning to inpaint does not bring much improvement to the absolute camera pose estimation.

In Fig. 7, we compare the results of NeurHal against state-of-the-art local feature matching methods. In low-overlap settings, very few keypoints' correspondents can be identified and many keypoints' correspondents have to be outpainted. In this case, we find that NeurHal is able to estimate the camera pose correctly significantly more often than any other method, since NeurHal is the only method able to outpaint correspondences (see Fig. 4). For high-overlap image pairs, the ability to

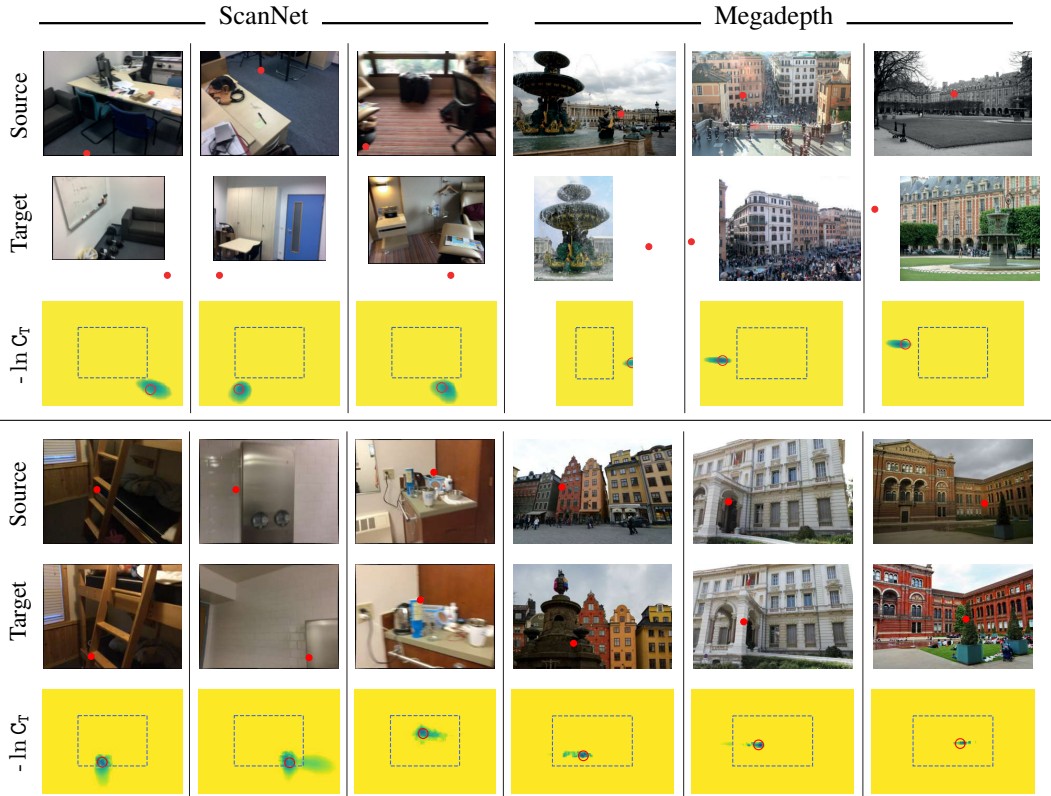

Figure 5: **Ability to hallucinate - Qualitative inpainting/outpainting results.** To illustrate the ability of NeurHal to hallucinate correspondents, we display correspondence maps predicted by NeurHal on image pairs (captured in scenes that were not seen at training-time): (top row) outpainting examples, (bottom row) inpainting examples. In the source image, the red dot is a keypoint. In the target image and in the (negative-log) correspondence map, the red dot represents the ground truth keypoint's correspondent. The dashed rectangles represent the borders of the target images. More results on the NYU and ETH-3D datasets can be found in the appendix D.1.

hallucinate is not useful since many keypoints' correspondents can be identified. In this case, we find that state-of-the-art local feature matching methods to be slightly better than NeurHal. This is likely due to the fact that NeurHal outputs low resolution correspondences maps while the other methods output high resolution correspondences. The overall performance shows that NeurHal significantly outperforms all the competitors, which allows us to conclude that the ability of NeurHal to outpaint correspondences is beneficial for absolute pose estimation. Technical details concerning the previous experiment as well as additional experiments concerning the application to absolute camera pose estimation are provided in Sec. B).

## 5 LIMITATIONS

We identified the following limitations for our approach: $(i)$ - The previous experiments showed that NeurHal is able to inpaint correspondences but the inpainted correspondence maps are much less peaked compared to the outpainted correspondence maps. This is likely due to the fact that inpainting correspondences is much more difficult than outpainting correspondences (see Sec 3.1). $(ii)$ - The proposed architecture outputs low resolution correspondence maps (see Sec. 3.3), *e.g.* $160 \times 120$ for input images of size $640 \times 480$ and an amount of padding $\gamma = 50\%$. This is essentially due to the quadratic complexity of attention layers we use (see Sec. C.1 of the appendix). $(iii)$ - Our approach is able to outpaint correspondences but our correspondence maps have a finite size. Thus, in the case where a keypoint's correspondent falls outside the correspondence map, the resulting correspondence map would be erroneous. We believe these three limitations are interesting future research directions.

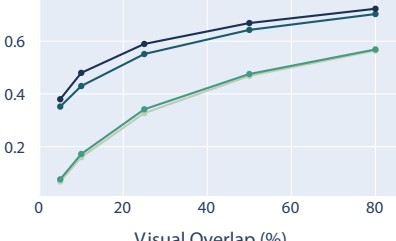

| Method | Training correspondences | | |
|---|---|---|---|
| | Identified | Inpainted | Outpainted |
| | ✓ | | |
| | ✓ | ✓ | |
| | ✓ | | ✓ |
| | ✓ | ✓ | ✓ |

Figure 6: **Ablation study - Impact of learning to hallucinate for absolute camera pose estimation.** We compare the influence of adding inpainting and outpainting ($\gamma = 50\%$) tasks when training NeurHal. We report the percentage of camera poses being correctly estimated for image pairs having an overlap between $2\%$ and $x\%$, as a function of $x$, on ScanNet (Dai et al., 2017), with thresholds for translation and rotation errors of $\tau_t = 1.5m$ and $\tau_r = 20.0°$. Learning to hallucinate correspondences (especially outpainting) significantly improves the amount of correctly estimated poses.

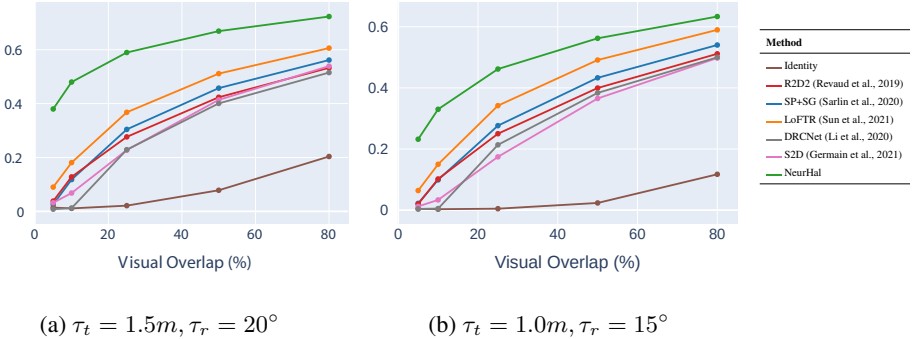

(a) $\tau_t = 1.5m, \tau_r = 20°$        (b) $\tau_t = 1.0m, \tau_r = 15°$

Figure 7: **Absolute camera pose experiment.** We compare the performance of NeurHal against state-of-the-art local feature matching methods on ScanNet (Dai et al., 2017). The "identity" method consists in systematically predicting the identity pose. We report the percentage of camera poses being correctly estimated for pairs of images that have an overlap between $2\%$ and $x\%$, as a function of $x$, for two rotation and translation error thresholds. See discussion in Sec. 4.2.

## 6 CONCLUSION

To the best of our knowledge, this paper is the first attempt to learn to inpaint and outpaint correspondences. We proposed an analysis of this novel learning task, which has guided us towards employing an appropriate loss function and designing the architecture of our network. We experimentally demonstrated that our network is indeed able to inpaint and outpaint correspondences on pairs of images captured in scenes that were not seen at training-time, in both indoor (ScanNet) and outdoor (Megadepth) settings. We also tested our network on other datasets (ETH3D and NYU) and discovered that our model has strong generalization ability. We then tried to experimentally illustrate that hallucinating correspondences is not just a fundamental AI problem but is also interesting from a practical point of view. We applied our network to an absolute camera pose estimation problem and found that hallucinating correspondences, especially outpainting correspondences, allowed to significantly outperform the state-of-the-art feature matching methods in terms of robustness of the resulting pose estimate. Beyond this absolute pose estimation application, this work points to new research directions such as integrating correspondence hallucination into Structure-from-Motion pipelines to make them more robust when few images are available.

## 7 ETHICS STATEMENT

The method described in this paper has the potential to greatly improve many computer vision-based industrial applications, especially those involving visual localization in GPS-denied or cluttered environments. For example robotics or augmented reality applications could benefit from our algorithm to better relocalize within their surroundings, which could lead to more reliable and overall safer behaviours. If this was to be applied to autonomous driving or drone-based search and rescue, one could appreciate the positive societal impact of our method. On the other hand like many computer vision algorithms, it could be applied to improve robustness of malicious devices such as weaponized UAVs, or invade citizens privacy through environment re-identification. Thankfully as AI technology advances, discussions and regulations are brought forward by governments and public entities.

These ethical debates pave the way for a brighter future and can only make us think NeurHal will more bring benefits than harms to society.

## 8 REPRODUCIBILITY

We provide the NeurHal model architecture and weights in the supplementary material. We also release a simple evaluation script that generates qualitative results, and show in a notebook the results obtained on an image pair captured indoors using a smartphone.

## ACKNOWLEDGEMENT

The authors would like to thank Matthieu Vilain and Rémi Giraud for their insight on visual correspondence hallucination. This project has received funding from the Bosch Research Foundation (*Bosch Forschungsstiftung*). This work was granted access to the HPC resources of IDRIS under the allocation 2021-AD011011682R1 made by GENCI.

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

APPENDIX

In the following pages, we present additional experiments and technical details about our visual correspondence hallucination method NeurHal. We present additional experiments on the ability to hallucinate in Sec. A and on camera pose estimation in Sec. B. We describe technical details in Sec. C and provide additional qualitative results in Sec. D.

## A   ADDITIONAL EXPERIMENTS CONCERNING THE ABILITY TO HALLUCINATE CORRESPONDENCES

In this section we first present an additional ablation study on the ability to hallucinate, followed by additional insights on our model internal functioning.

### A.1   IMPACT OF LEARNING TO INPAINT AND OUTPAINT

To supplement the study made in Sec. 4.1, we now aim at evaluating the impact of learning to inpaint and outpaint specifically. To do so, we isolate keypoints with the *identified*, *inpainted* and *outpainted* labels in our ScanNet (Dai et al., 2017) evaluation set.

In Fig. 8, we show the results of an ablation study on NeurHal's training setup. We report for the identification, inpainting and outpainting tasks two sets of cumulative histograms: 1) the NRE costs at ground truth keypoint correspondents' locations, and 2) distances between the argmax of the correspondence map and the ground truth location. On NRE cost cumulative histograms, we also report the results from the uniform distribution, for models trained both with and without outpainting ($\gamma = 0\%$ and $\gamma = 50\%$ respectively).

For the identification task (Fig. 8 (a)) we find that all methods yield a consistent performance. The left figure reveals that NeurHal predictions are significantly above the uniform distribution, indicating peaky maps and thus confident predictions. The right figure shows that the distance of the argmax location *w.r.t.* the ground truth is also robust (NeurHal predicts at 1/8th of the original resolution but the histogram is computed at full resolution).

For the inpainting task (Fig. 8 (b)) we can draw similar conclusions. We find however that correspondence maps are overall less peaky and closer to their respective uniform distribution, which indicates that predictions are less confident. We also find that even though it was not trained to inpaint, the identification baseline is surprisingly able to inpaint correspondences as its performance is not far from the identification+inpainting model.

Lastly for the outpainting task (Fig. 8(c)), we find that learning to outpaint gives a significant boost in performance on both the NRE distribution and correspondents locations. We also find that jointly learning to inpaint and outpaint is beneficial to the quality of the outpainted cost maps, which implies that both objectives are complementary.

## A.2 ABILITY TO HALLUCINATE: NEURHAL HALLUCINATION VS. HOMOGRAPHY-BASED WARPING

To compare the ability to hallucinate of NeurHal against a non learning-based approach, we report in Fig. 9 the performance of homography-based warping approaches that do no rely on correspondence hallucination.

We compare the performance of a) NeurHal trained to both identify and hallucinate correspondences against b) a version of NeurHal trained without correspondence hallucination followed by a homography-based warping stage to hallucinate correspondences. We derive several baselines of b). We report the performance obtained using a simple least-squares solver from all predicted correspondences, and a RANSAC alternative. We also report the performance of using an oracle prior to the homography estimation stage to filter out outlier correspondence predictions. We lastly report the performance of estimating the homography using ground-truth identifiable correspondences inside a RANSAC loop. For completeness we show the performance of RANSAC-based homography estimation for several inlier thresholds. In all cases, we find that performing correspondence hallucination using NeurHal significantly outperforms all homography-based alternatives that do not resort to ground-truth information.

This can be attributed to the lack of sufficient visual overlap between image pairs that prevent from obtaining an accurate homography estimate, as well as the planar-assumption of this model.

Interestingly, it can be seen that with perfect correspondences inpainted correspondences can be accurately recovered. On the other hand, outpainted correspondences do not seem to be robustly retrievable through a simple homography estimation.

## A.3 ADDITIONAL INSIGHTS ON CORRESPONDENCE HALLUCINATION

While it is tempting to draw hypotheses regarding the internal functioning of NeurHal, we would like to highlight that this should be done with great care. Indeed, the sheer complexity of the operations run in the many attention layers of NeurHal prevents from interpreting the reasoning that leads to the outputted correspondence maps. From a higher level however, we can reasonably speculate that our model implicitly learns to jointly predict the depth of the source image and the relative camera pose to warp the source keypoints and hallucinate their correspondents in the target image.

## A.4 INPAINTED VS. OUTPAINTED CORRESPONDENCE MAPS

We can observe that the inpainted correspondence maps are less peaked than the outpainted correspondence maps (see Fig. 3). We believe this is because outpainting a correspondent essentially consists in transferring the features from location $p_{s,i}$ in the source features to location $p_{t,i}$ in the target features to obtain $d_{s,i} = D_{T,pad}(p_{t,i})$. In the inpainting case, $p_{s,i}$ is occluded by an object in the target image, and this object is (often) also visible in the source image at location $p_{s,occ}$. Thus, to produce peaked correspondence maps for both $p_s$ and $p_{s,occ}$, the network has to output features such that $d_{s,i} = D_{T,pad}(p_{t,i}) = d_{s,occ}$ which is more difficult than just $d_{s,i} = D_{T,pad}(p_{t,i})$

## A.5 ABILITY TO HALLUCINATE: NEURHAL VS. (GERMAIN ET AL., 2021)

The architecture proposed by Germain et al. (2021) is not able to predict outpainted correspondences. They do consider an extra category for un-matched keypoints but the probability for that category is

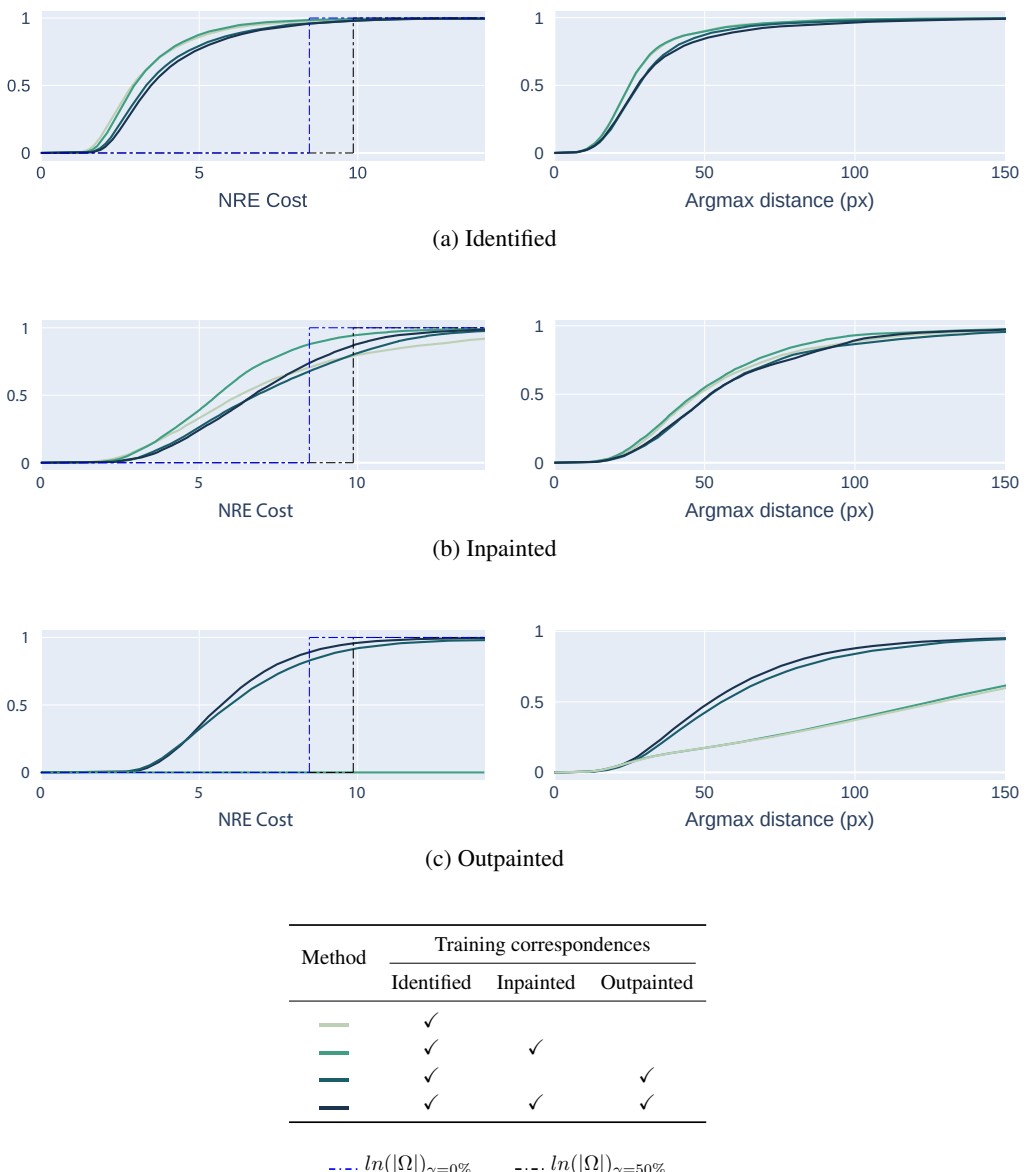

Figure 8: **Ability to hallucinate - Ablation study on ScanNet.** We compare the influence of adding inpainting and outpainting when training NeurHal. **(left column)** We report the percentage of keypoint's correspondents whose NRE cost is lower than $x$, as a function of $x$, for (a) identified (b) inpainted and (c) outpainted keypoints. **(right column)** We report the percentage of keypoint's correspondents whose distance *w.r.t.* the ground truth is lower than $x$ pixels, as a function of $x$, for the same categories.

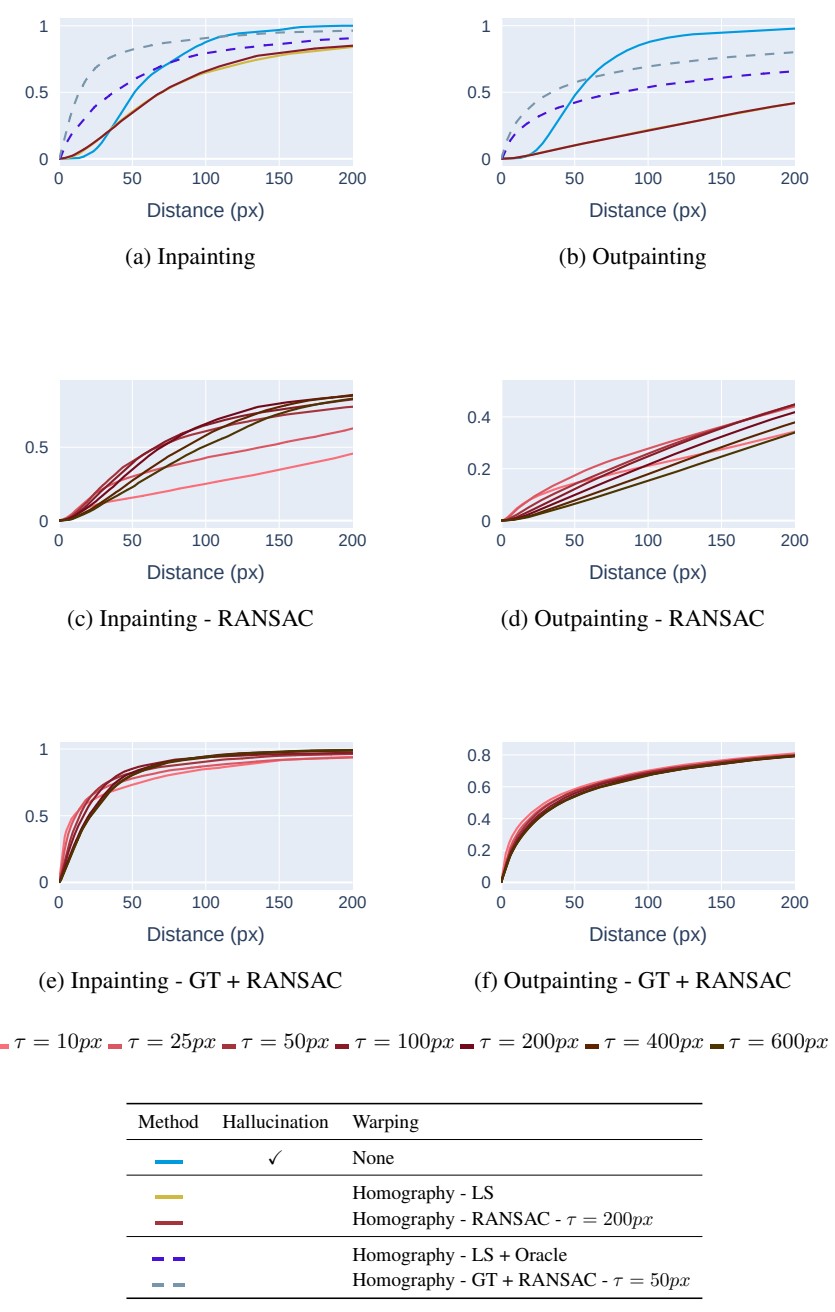

Figure 9: **Ability to hallucinate - Homography-based warping:** We compare the performance of 1) NeurHal trained to both identify and hallucinate correspondences against 2) a version of NeurHal trained without correspondence hallucination followed by a homography-based warping stage to hallucinate correspondences and 3) a homography estimated from the ground truth (GT) identified correspondences. We report the performance to predict $(a)$ inpainted and $(b)$ outpainted correspondents locations. For each method we report the percentage of keypoint's correspondents whose distance *w.r.t.* the ground truth location is lower than $x$ pixels, as a function of $x$. We show in $(c)$ and $(d)$ the performance of RANSAC-based homography estimation for various inlier thresholds. We show in $(e)$ and $(f)$ similar curves when using ground-truth correspondences to estimate the homography. See text for more details.

set to zero, ie. the network does not output any score for that category. Setting this probability to zero is due to the fact that (Germain et al., 2021) considers a classical siamese CNN architecture that does not allow the features of both images to communicate. (Germain et al., 2021) is what we called, in the introduction, a "pure pattern recognition approach". Moreover, even if (Germain et al., 2021) were using a non-siamese architecture, their method would output a single score for the category "un-matched keypoint" which would allow the network to detect when the correspondent is not visible but would not be sufficient to outpaint the location of the correspondent.

## B  ADDITIONAL EXPERIMENTS CONCERNING THE APPLICATION TO CAMERA POSE ESTIMATION

In this section, we present additional experiments on correspondence hallucination for camera pose estimation. We begin with a study on the impact of the pose estimator in Sec. B.1, followed by a study on the impact of the padding value $\gamma$ in Sec. B.2. Lastly, we present in Sec. B.4 additional results on indoor camera pose estimation.

### B.1  INFLUENCE OF THE POSE ESTIMATOR: (GERMAIN ET AL., 2021) VS. (CHUM ET AL., 2003)

(Germain et al., 2021) provides a pose estimation framework which leverages dense keypoint matching uncertainties to predict more accurate and robust camera poses. Compared to the standard pose estimator presented in (Chum et al., 2003) which relies on sparse 2D-to-3D correspondences, the method from (Germain et al., 2021) preserves rich information in the form of dense loss maps that is particularly suited for ambiguous matches. For the problem of correspondence hallucination we find the loss maps of both outpainted and inpainted correspondences are usually unimodal but quite diffuse, and are thus particularly suited for this pose estimator.

To study the influence of the pose estimator, we report in Fig. 10 the performance of NeurHal + (Germain et al., 2021) vs. NeurHal + (Chum et al., 2003). To estimate the camera pose using the method presented in (Chum et al., 2003), we simply take the argmax of each correspondence map and treat it as a sparse 2D correspondent in the query image. We also include the performance of NeurHal when trained without visual correspondence hallucination (*i.e.* trained using only identified ground truth correspondences.)

We find that the two methods trained without hallucination have poor performances for very low-overlap image pairs which underlines the importance of correspondence hallucination in such cases.

Concerning NeurHal trained with hallucination and using the pose estimator (Chum et al., 2003), taking the argmax of a very coarse correspondence map prevents the pose estimator from achieving good results.

NeurHal trained with hallucination and coupled with the pose estimator of Germain et al. (2021) achieves the best results which shows that to obtain robust absolute camera estimates it is important to *combine* the ability to hallucinate correspondences of NeurHal with the pose estimator from (Germain et al., 2021).

### B.2  IMPACT OF THE VALUE OF $\gamma$

We report in Fig. 11 the absolute camera pose estimation performance for varying values of $\gamma$. We compute the percentage of camera poses being correctly estimated for ScanNet (Dai et al., 2017) test images pairs that have an overlap between $2\%$ and $x\%$ (as a function of $x$) for a translation threshold of $1.5m$ and a rotation threshold of $20.0°$.

We find that using only a small percentage of outpainting such as $\gamma = 10\%$ does not improve the performance which is most likely due to the small amount of added training keypoints. For higher $\gamma$ values however significant gains are visible, especially at small visual overlaps. This experiment demonstrates the benefit of learning to outpaint correspondences beyond image borders, and broaden the extent of usable source keypoints to perform camera pose estimation.

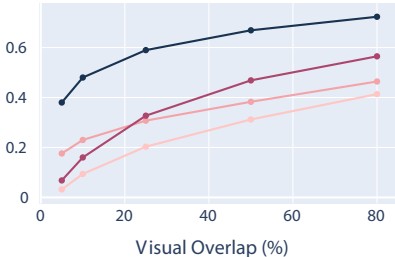

| Method | Hallucination | Pose estimator |
|---|---|---|
| | | (Chum et al., 2003) |
| | ✓ | (Chum et al., 2003) |
| | | (Germain et al., 2021) |
| | ✓ | (Germain et al., 2021) |

Figure 10: **Influence of the pose estimator: (Germain et al., 2021) vs. (Chum et al., 2003):** To study the influence of using the pose estimator proposed in (Germain et al., 2021) compared to using the pose estimator from (Chum et al., 2003), we report the performance of NeurHal with both estimators. We also include, for both estimators, the performance of NeurHal trained with identified correspondences only (*i.e.* without hallucination). We report the percentage of camera poses being correctly estimated for pairs of ScanNet (Dai et al., 2017) images that have an overlap between 2% and $x\%$ (as a function of $x$).

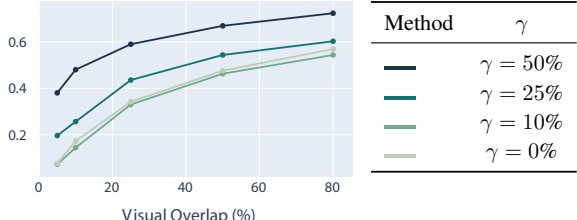

| Method | $\gamma$ |
|---|---|
| | $\gamma = 50\%$ |
| | $\gamma = 25\%$ |
| | $\gamma = 10\%$ |
| | $\gamma = 0\%$ |

Figure 11: **Impact of the value of $\gamma$:** For increasing values of $\gamma$, we report the percentage of camera poses being correctly estimated for pairs of ScanNet images that have an overlap between 2% and $x\%$ (as a function of $x$), for $\tau_t = 1.5m$ and $\tau_r = 20.0°$. We find that a small value of $\gamma = 10\%$ yields no benefit and even damages performance, while values of $\gamma = 25\%$ and $\gamma = 50\%$ bring significant improvements, especially at small visual overlaps.

We report in Fig. 12 the camera field-of-view as a function of the padding parameter. We find that $\gamma = 50\%$ provides 130° and 71° of field-of-view on average on ScanNet and Megadepth respectively, which is significantly wider than $\gamma = 0\%$.

### B.3 ANALYSIS OF THE IMPACT OF INPAINTING AND OUTPAINTING

In Fig. 11 we reported the percentage of camera poses being correctly estimated for several values of $\gamma$, which demonstrates the benefits of outpainting with a large $\gamma$ for camera pose estimation. In Fig. 6 we also showed that learning to inpaint does not bring any significant improvement. We believe that outpainting improves the camera pose because outpainted correspondences are outside the field of view and thus complement the identified correspondences, and thus better constrain the camera pose estimate. On the contrary, inpainted correspondences are usually surrounded by identified correspondences, thus the information they provide is redundant and does not allow to better constrain the camera pose estimate.

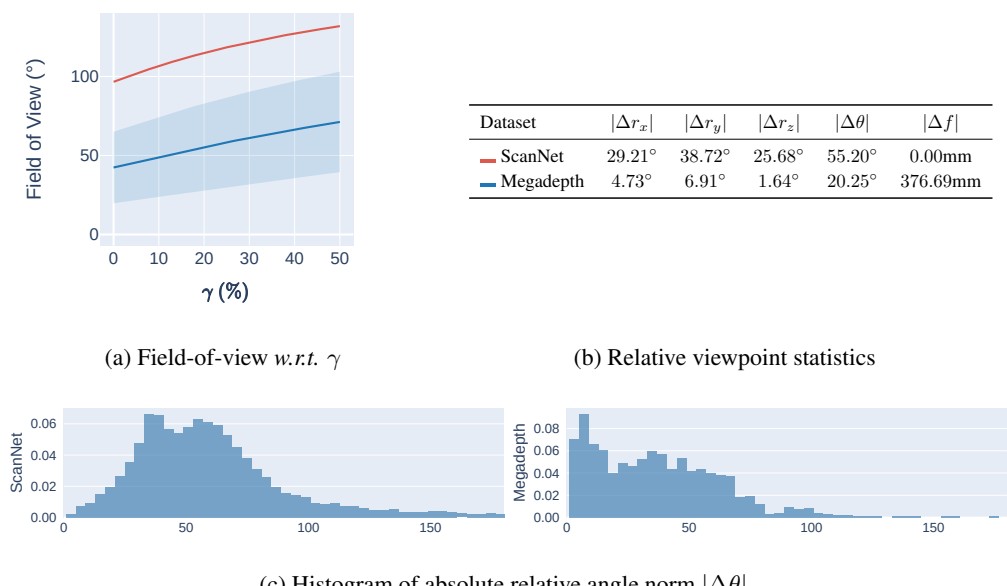

(a) Field-of-view *w.r.t.* $\gamma$

(b) Relative viewpoint statistics

| Dataset | $|\Delta r_x|$ | $|\Delta r_y|$ | $|\Delta r_z|$ | $|\Delta\theta|$ | $|\Delta f|$ |
|---|---|---|---|---|---|
| ScanNet | 29.21° | 38.72° | 25.68° | 55.20° | 0.00mm |
| Megadepth | 4.73° | 6.91° | 1.64° | 20.25° | 376.69mm |

(c) Histogram of absolute relative angle norm $|\Delta\theta|$

Figure 12: **Field-of-view as a function of $\gamma$ and relative viewpoint statistics:** We report in $(a)$ the average camera field-of-view as a function of $\gamma$ on ScanNet (Dai et al., 2017) and Megadepth (Li & Snavely, 2018) images. We find that $\gamma = 50\%$ enables a significant amount of additional visual content to reproject within the image boundaries. We report in $(b)$ the median absolute difference in rotation along the $x$, $y$ and $z$ axis, norm of the relative rotation, along with the difference in focal length on low-overlap image pairs for ScanNet (Dai et al., 2017) and Megadepth (Li & Snavely, 2018). We report in (c) the histogram of absolute relative angle norm on both datasets. We find ScanNet image pairs exhibit strong relative angular motion while Megadepth image pairs display predominantly zoom-ins and zoom-outs.

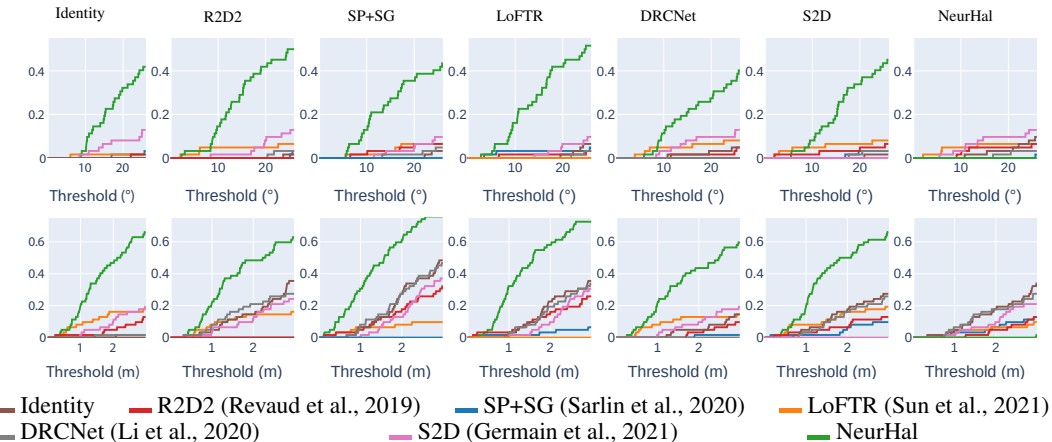

— Identity    — R2D2 (Revaud et al., 2019)    — SP+SG (Sarlin et al., 2020)    — LoFTR (Sun et al., 2021)
— DRCNet (Li et al., 2020)    — S2D (Germain et al., 2021)    — NeurHal

Figure 13: **Camera pose estimation experiment - Worst cases:** We report the performance of NeurHal and state-of-the-art feature matching methods on ScanNet (Dai et al., 2017) image pairs with visual overlaps between 2% and 5%. For every column, we subselect the 25% of images pairs with the worst predictions for a given method. We find that in all cases, NeurHal strongly outperforms its competitors. On the contrary, on the worst NeurHal predictions state-of-the-art methods achieve a much lower performance, which is either on par or lower than the predictions obtained using the Identity.

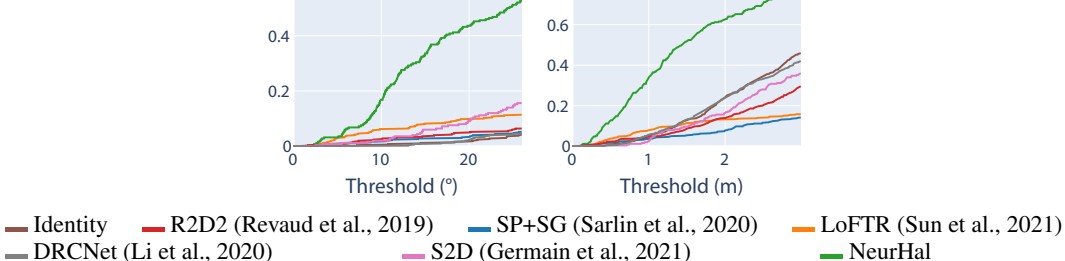

— Identity    — R2D2 (Revaud et al., 2019)    — SP+SG (Sarlin et al., 2020)    — LoFTR (Sun et al., 2021)
— DRCNet (Li et al., 2020)    — S2D (Germain et al., 2021)    — NeurHal

Figure 14: **Camera pose estimation experiment - varying the threshold values:** We report the performance of NeurHal and state-of-the-art feature matching methods on ScanNet (Dai et al., 2017) image pairs with visual overlaps between 2% and 5%. For various angular and translation thresholds we report the percentage of correctly localized images. We find that in all cases, NeurHal strongly outperforms its competitors.

## B.4 ADDITIONAL INDOOR POSE ESTIMATION RESULTS

In addition to the results presented in Fig. 7, we report in Fig. 13 the performance of NeurHal and state-of-the-art feature matching methods on ScanNet (Dai et al., 2017) image pairs with visual overlaps between 2% and 5%. For every method, we subselect the 25% of images pairs with the worst predictions, and compare it with the performance of its competitors. We find that in all cases, NeurHal strongly outperforms its competitors. On the worst NeurHal predictions, state-of-the-art methods achieve a much lower performance. For this category we can observe that all NeurHal competitors are either on par or achieve a lower performance than the Identity predictions.

This figure highlights the fact that when NeurHal fails to correctly estimate the camera pose, all the competitors also fail since all the methods perform similarly to the "identity" method, *i.e.* the method that consists in systematically predicting the identity pose.

Fig. 14 shows that NeurHal is much more robust than state-of-the-art local feature matching methods for pairs of images with a low overlap.

## C    TECHNICAL DETAILS

### C.1    ARCHITECTURE DETAILS

NeurHal's architecture can be separated in two building blocks: the convolutional backbone and the multi-head attention block.

**Convolutional backbone.**    The convolutional backbone consists of a truncated Inceptionv3 (Szegedy et al., 2016) model (up to Mixed-6a, 768-dimensional descriptors), modified as per Germain et al. (2021) to provide, in the case of ScanNet (Dai et al., 2017), a $1/8$ output-to-input resolution ratio. To help with memory consumption we apply a simple 2D convolutional layer to compress the descriptor size to 384. In the case where $\gamma > 0$, we subsequently pad $H_T$ with the learned vector $\boldsymbol{\lambda}$, producing $H_{T,\text{pad}}$.

**Positional encoding.**    After computing $H_S$ and $H_{T,\text{pad}}$ with the convolutional backbone, positional encoding is applied to both dense feature maps. Similarly to SuperGlue (Sarlin et al., 2020), we use a 6-layer MLP of size (32, 64, 128, 256, 384), mapping a positional meshgrid between $(-1, 1)$ (centered around the image center) to higher dimensionalities. BatchNorm and ReLU layers are placed between every module. In our experiments, we tried adding more positional encoding layers but found it did not make a difference in performance. After applying the positional encoding, sparse descriptors $\{\mathbf{d}_{S,n}\}_{n=1...N}$ are bilinearly interpolated at $\{\mathbf{p}_{S,n}\}_{n=1...N}$ in $H_S$.

**Self-attention.**    Following the positional encoding, a single multi-head attention layer is applied on $H_{T,\text{pad}}$, with 4 heads. It consists of a standard dot-product attention (Vaswani et al., 2017), coupled with a gating mechanism. For a given query $Q$, key $K$ and value $V$, we compute the attention as $\text{Attention}(Q, K, V) = \text{softmax}(g * QK^T)V$ where $g = \sigma(max(QK))$. To mitigate the quadratic cost of the dot-product attention, we also apply a max-pooling operator on keys and values with a stride of 2, as we empirically found it had very little impact on performance. We also tried using a Linear Transformer (*e.g.* LinFormer (Katharopoulos et al., 2020)) architecture, but despite trying numerous variants we found it consistently damaged the convergence of the model.

**Cross-attention.**    Using the same attention-layer design, we subsequently apply it once between $\{\mathbf{d}_{S,n}\}_{n=1...N}$ and $H_S$. This layer allows for communication between the interpolated source descriptors which will be used to produce the final correspondence maps, and the original dense source image content. Then, we apply $k$ cross-attention layers between $\{\mathbf{d}_{S,n}\}_{n=1...N}$ and $H_{T,\text{pad}}$. We empirically found these layers to be most important, as they allow for direct communication between the sparse source descriptors and the dense target feature maps, prior to the correspondence maps computation. After trying different values for $k$ and with memory consumption in mind, we settled for $k = 4$ in all our experiments.

**Affine transformation.**    The affine transformation matrix $K_C$ for the correspondence map $C_T$ of resolution $(W_T, H_T)$ is computed from the target image calibration matrix $K_T$ and downscaling factors $s$ using:

$$K_C = \begin{bmatrix} s & 0 & \frac{s-1}{2} \\ 0 & s & \frac{s-1}{2} \\ 0 & 0 & 1 \end{bmatrix} K_T + \begin{bmatrix} 0 & 0 & \gamma W_T \\ 0 & 0 & \gamma H_T \\ 0 & 0 & 0 \end{bmatrix} \tag{2}$$

We can determine if a point lies within the boundaries of $C_T$ if its $(x, y)$ coordinates are between $(-\gamma W_T, -\gamma H_T)$ and $((1 + \gamma)W_T, (1 + \gamma)H_T)$.

**Implementation.**    The model is implemented in PyTorch (Paszke et al., 2017). For an indoor sample with 2000 keypoints it has an average throughput of $8.84$ image/s on an NVIDIA RTX 3070 GPU. We report the number of parameters in our model in Table 1.

| Layer | # of parameters |
|---|---|
| CNN | 2.4 M |
| Positional Encoding | 142 K |
| Self-Attention | 1.9 M |
| Cross-Attention | 7.2 M |
| Total | 11.7 M |

Table 1: **Number of parameters in NeurHal**

## C.2 DATASETS AND TRAINING DETAILS

**ScanNet.** The ScanNet (Dai et al., 2017) dataset is a large-scale indoor dataset containing monocular RGB videos and dense depth images, along with ground truth absolute camera poses. As SuperGlue (Sarlin et al., 2020) and LoFTR (Sun et al., 2021), we pre-compute the visual overlaps between all image pairs for both training and test scenes. For the training set we sample images with a visual overlap between 2% and 50% from the ScanNet training scenes, which provides us with challenging images to handle. We assemble $6M$ image pairs and randomly subsample $200k$ pairs at every training epoch. For testing images, we sample $2,500$ image pairs with overlaps between 2% and 80% from the ScanNet testing scenes, using several bins to ensure the sampling is close to being uniform. For both training and testing images, we sample keypoints in the source image along a regular grid with cell sizes of 16 pixels. We remove keypoints with invalid depth, as well as those where the local depth gradient is too high, as the depth information might not be reliable. We mark keypoints falling outside the target image plane as being outpainted, and we automatically detect the keypoints to inpaint through a cyclic projection of the source keypoints to the target image and back. The remaining keypoints are labeled as identifiable. For all ScanNet experiments, NeurHal uses a $1/8$ output-to-input resolution ratio, with a target correspondence map maximum edge size of 80 pixels (when $\gamma = 0\%$).

**Megadepth.** We use Megadepth (Li & Snavely, 2018) to train and evaluate NeurHal on outdoor images. This dataset contains over one million images captured in touristic places, and split in 196 scenes. To train NeurHal and following Germain et al. (2021) guidelines, we use the provided SIFT (Lowe, 2004)-based 3D reconstruction which was made with COLMAP (Schönberger & Frahm, 2016). Because the sparse 3D point cloud comes from SfM, we find however that very little keypoints can be marked as inpainted. Indeed, no 3D reconstruction is applied to objects or people occluding the scene. To allow for a wide variety of image pairs we use the sparse reconstruction to estimate the visual overlap and sample pairs with an overlap between 20% and 100%. We however find this overlap estimation to be quite unreliable, as only part of the scene is usually reconstructed. Since Megadepth (Li & Snavely, 2018) images are of much higher resolution than ScanNet (Dai et al., 2017), we configure NeurHal to use a $1/16$ output-to-input resolution (with a simple max-pooling layer in the CNN). We set the target correspondence map maximum edge size of 60 pixels (when $\gamma = 0\%$), to allow for space in memory when $\gamma = 50\%$.

**Overlap estimation.** For a given pair of images, we approximate the visual overlap by computing the covisibility ratio of keypoints for every image pair. For a given source and target image pair, we first compute the source-to-target and target-to-source covisibility ratios using ground truth depth data and camera poses. We then define the visual overlap as the minimum between both ratios. On Megadepth we find this overlap estimation to be fairly noisy, as depth is only partially known.

**Optimizers and scheduling.** On both datasets NeurHal is trained for a maximum of 40 epochs. We use an initial learning rate of $10^{-3}$, with a linear learning rate warm-up in 3 epochs from 0.1 of the initial learning rate. As Sun et al. (2021), we decay the learning rate by 0.5 every 8 epochs starting from the 8th epoch. We apply the linear scaling rule and use a batch size of 8 over 8 NVIDIA V100 GPUs. We use the AdamW (Loshchilov & Hutter, 2019) optimizer, with a weight decay of 0.1. In all training procedures, we randomly initialize the model weights.

### C.3 EVALUATION DETAILS

**Evaluation protocol.**    All baselines follow the same standard protocol in which we: 1) Compute 2D-2D correspondences between the reference image and the query image, 2) Lift these 2D-2D correspondences to 2D-3D correspondences using the available 3D information for the reference image, 3) Estimate the camera pose given these 2D-3D correspondences by minimizing the Reprojection Error (RE), i.e. applying LO-RANSAC+PnP (Chum et al., 2003) followed by a non-linear iterative refinement. This approach is widely used and leads to state-of-the-art results in visual localization benchmarks. We also include results for Germain et al. (2021) which we call S2D. For the evaluation of Fig. 4, we find the inpainted and outpainted correspondents for LoFTR (Sun et al., 2021) and DRCNet (Li et al., 2020) by fetching the argmax 2D coordinates in the 4D matching confidence volume. For S2D and NeurHal, we simply take the argmax in correspondence maps for the same set of keypoints.

**Choice of threshold.**    We reported in Fig. 14 the performance of NeurHal, state-of-the-art feature matching methods and the identity pose, on ScanNet for several rotation and translation thresholds. We can see that arbitrarily choosing a threshold of $\tau_t = 1.5m$ and $\tau_r = 20.0°$ sets a hard objective as the identity pose is particularly poor.

**(Chum et al., 2003)-based pose estimator.**    For all (Chum et al., 2003)-based methods, we estimate the camera pose using the pycolmap python binding. We tune the RANSAC threshold for optimal performance, and mark all cases where less than 3 valid correspondences (*i.e.* with a valid depth value) as failure cases (infinite pose error). The remaining parameters are left as default. We follow the evaluation instructions provided by each method, and use indoor weights for SP+SG (Sarlin et al., 2020) and the dual-softmax indoor weights for LoFTR (Sun et al., 2021). In the case of NeurHal + (Chum et al., 2003), we simply read the argmax of the predicted correspondence maps to obtain explicit 2D-to-3D correspondences.

**(Germain et al., 2021)-based pose estimator.**    For both S2D (Germain et al., 2021) and NeurHal we only use coarse models, which operate at either 1/8th or 1/16th of the original input resolution. We first retrain the S2D coarse model (fully-convolutional Inceptionv3 (Szegedy et al., 2016), up to Mixed-6e) on the same training set as our method, with the same target resolution of 80 pixels. We refer to this model as S2D. Given correspondence maps and the depth map of the source image, we estimate the camera pose between the target image and the source image using the method proposed in Germain et al. (2021). For both S2D and NeurHal we use the same set of regularly sampled source keypoints (see Sec. C.1), and we perform camera pose estimation first using P3P inside an MSAC (Torr & Zisserman, 2000) loop. We run P3P for a maximum of $5,000$ iterations over the top-20% correspondences. We then apply a coarse GNC (Blake & Zisserman, 1987) over all source keypoints with $\sigma_{max} = 2.0$ and $\sigma_{min} = 0.6$. Let us highlight that in all the camera pose experiments, the performances of NeurHal are obtained by predicting *only* low resolution correspondence maps (see Sec. C.1).

## D ADDITIONAL QUALITATIVE RESULTS

### D.1 GENERALIZATION TO NEW DATASETS

So far we have demonstrated the ability of NeurHal to hallucinate correspondences on unseen validation scenes from both ScanNet (Dai et al., 2017) and Megadepth (Li & Snavely, 2018). In order to further demonstrate the generalization capacity of NeurHal, we report qualitative results obtained on the NYU Depth Dataset (Nathan Silberman & Fergus, 2012) in Fig. 16 and on the ETH-3D (Schöps et al., 2017) dataset in Fig. 15. We use the set of indoor weights for NYU (*i.e.* NeurHal trained on ScanNet) and outdoor weights for ETH-3D (*i.e.* NeurHal trained on MegaDepth). We report the overlayed and upsampled coarse truncated loss map computed following Germain et al. (2021) on low-overlap image pairs. We find that NeurHal is able to robustly outpaint correspondences despite little visual overlaps and strong relative camera motions. These visuals demonstrate the strong generalization ability of NeurHal.

### D.2 QUALITATIVE CORRESPONDENCE HALLUCINATION RESULTS AND FAILURE CASES

To further demonstrate the ability of NeurHal to perform visual correspondence hallucination, we report in Fig. 17 and Fig. 18 qualitative results on ScanNet (Dai et al., 2017) and Megadepth (Li & Snavely, 2018) respectively on scenes that were not seen at training-time. In the target image and in the (negative log) correspondence map, the red dot represents the ground truth keypoint's correspondent. The dashed rectangles represent the borders of the target images.

Let us recall that NeurHal outputs probability distributions (*a.k.a.* correspondence maps) *assuming the two input images are partially overlapping*. It is essential to keep this assumption in mind when looking at these qualitative results. For instance, concerning the example Fig. 17 (b) (middle), it is very difficult for our human visual system to be sure that the two images are actually overlapping, and consequently the network prediction seems to good to be true. However, if we *assume* that there is an overlap, we realize that it is actually possible to perform correspondence hallucination, by drawing out the two skirting boards, to correctly outpaint the correspondent.

In fact, this overlapping assumption has a regularization effect in cases where the covisible image areas show no distinctive regions, and one image could be at an infinite translation of the other, *e.g.* Fig. 17 (b) (second to last).

In Fig. 17 (d) and Fig. 18 (d) we show failure cases where the correspondence maps modes predicted by NeurHal are either partially or completely off. We find that failure cases often correlate with strongly ambiguous image pairs, or images that have extremely limited visual overlap.

### D.3 QUALITATIVE CAMERA POSE ESTIMATION RESULTS

We show in Fig. 19 qualitative results in camera pose estimation on low-overlap images from ScanNet (Dai et al., 2017), for NeurHal and its three best-performing competitors. For every method we display the keypoints used as input to the camera pose estimator in the source image, along with their reprojection at the estimated camera pose in the target image. For methods using the pose estimator from (Chum et al., 2003), the keypoints are those that have been successfully matched. When using the pose estimator of Germain et al. (2021), the keypoints are those involved in the prediction of the dense NRE maps. We color in keypoints based on their spatial 2D position in the source image. We find that NeurHal strongly benefits from its outpainting ability, in comparison with all other competitors which struggle to find both sufficient and reliable correspondences. We also report in Fig. 20 failure cases for NeurHal. We find that such cases correspond to image pairs exhibiting extremely limited visual overlap, strong camera pose rotations and overall significant ambiguities.

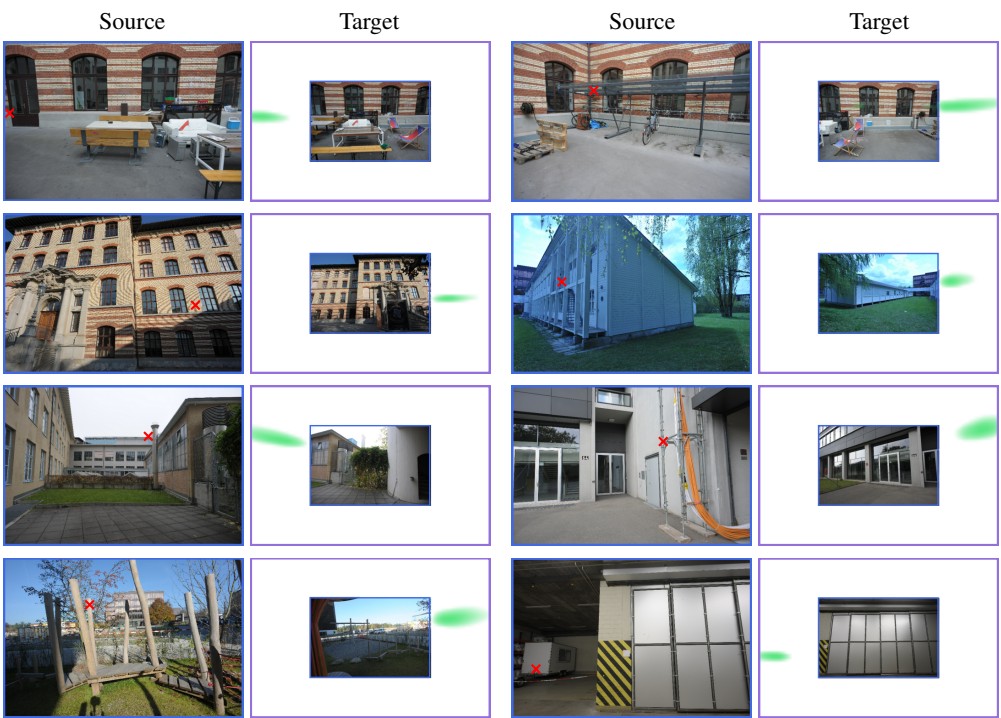

Figure 15: **Qualitative results on the ETH3D dataset:** We evaluate NeurHal on outdoor image pairs from the ETH-3D (Schöps et al., 2017) dataset and find it is able to outpaint correspondences despite low visual overlaps. We report pairs of source and target images and overlay the upsampled coarse loss map corresponding to the source detection (in red) on the target image.

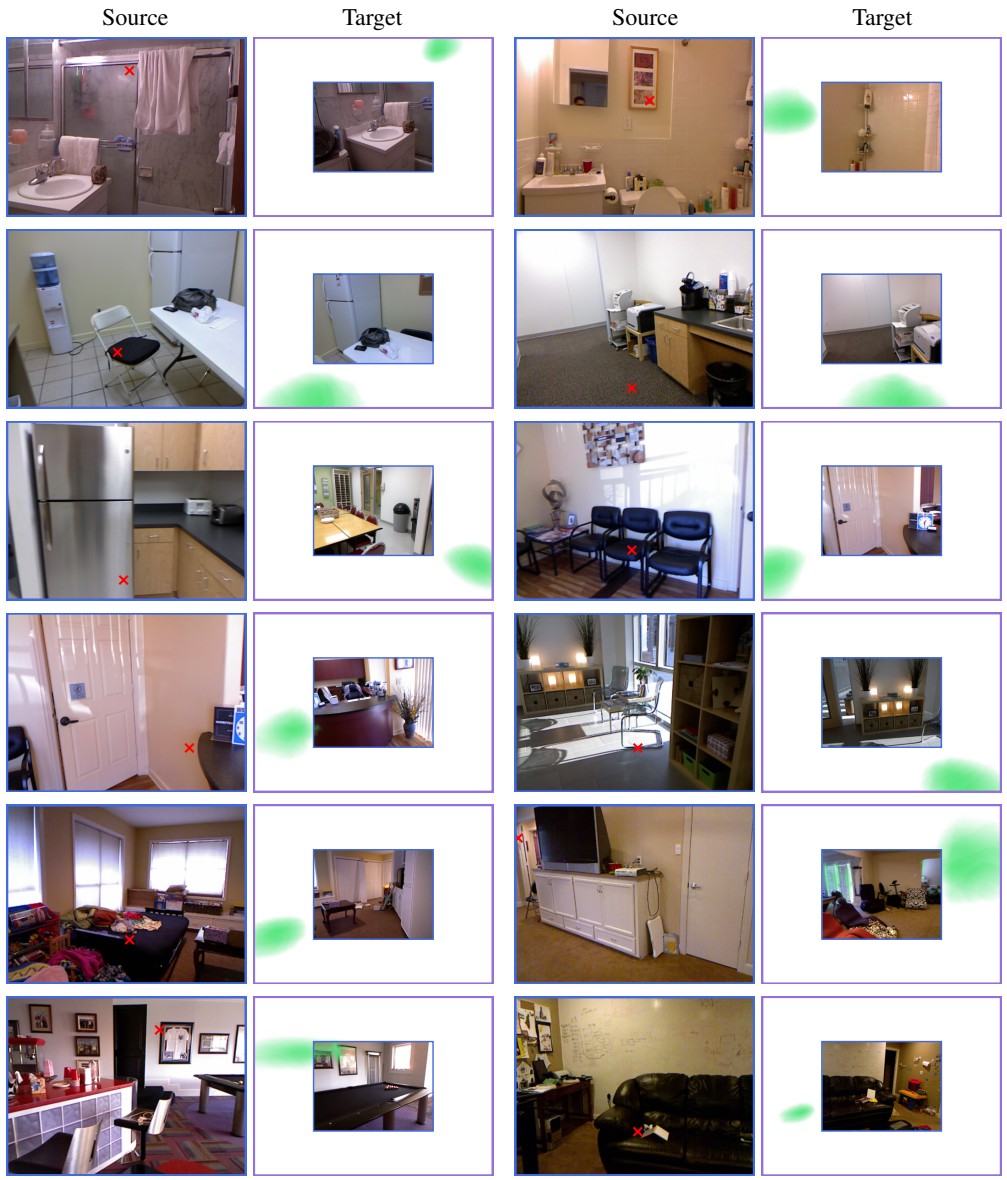

Figure 16: **Qualitative results on the NYU dataset:** We evaluate NeurHal on indoor images from the NYU (Nathan Silberman & Fergus, 2012) dataset and find it is able to outpaint correspondences despite low visual overlaps. We report pairs of source and target images and overlay the upsampled coarse loss map corresponding to the source detection (in red) on the target image.

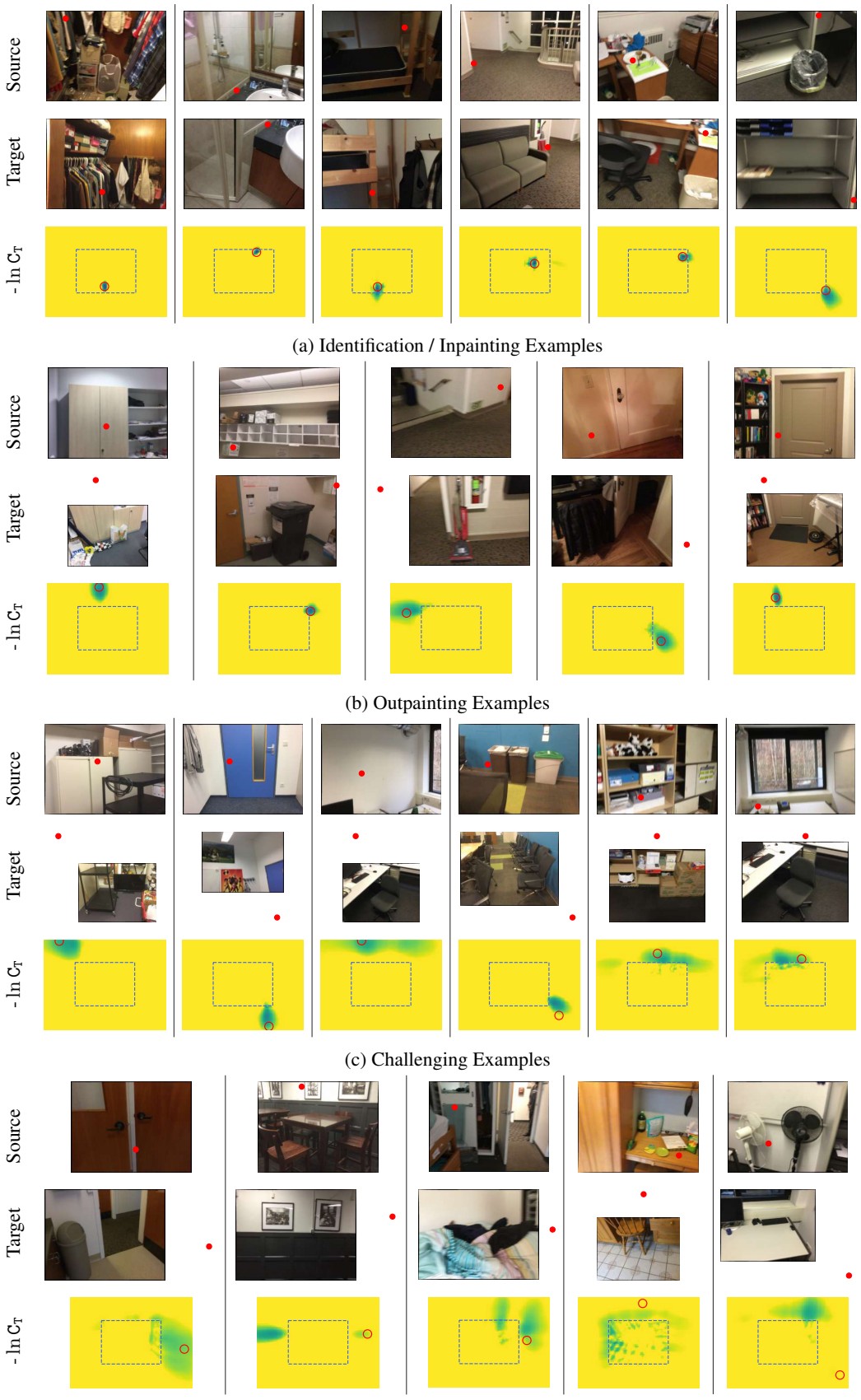

Figure 17: **Additional qualitative ScanNet (Dai et al., 2017) examples.** See text for details.

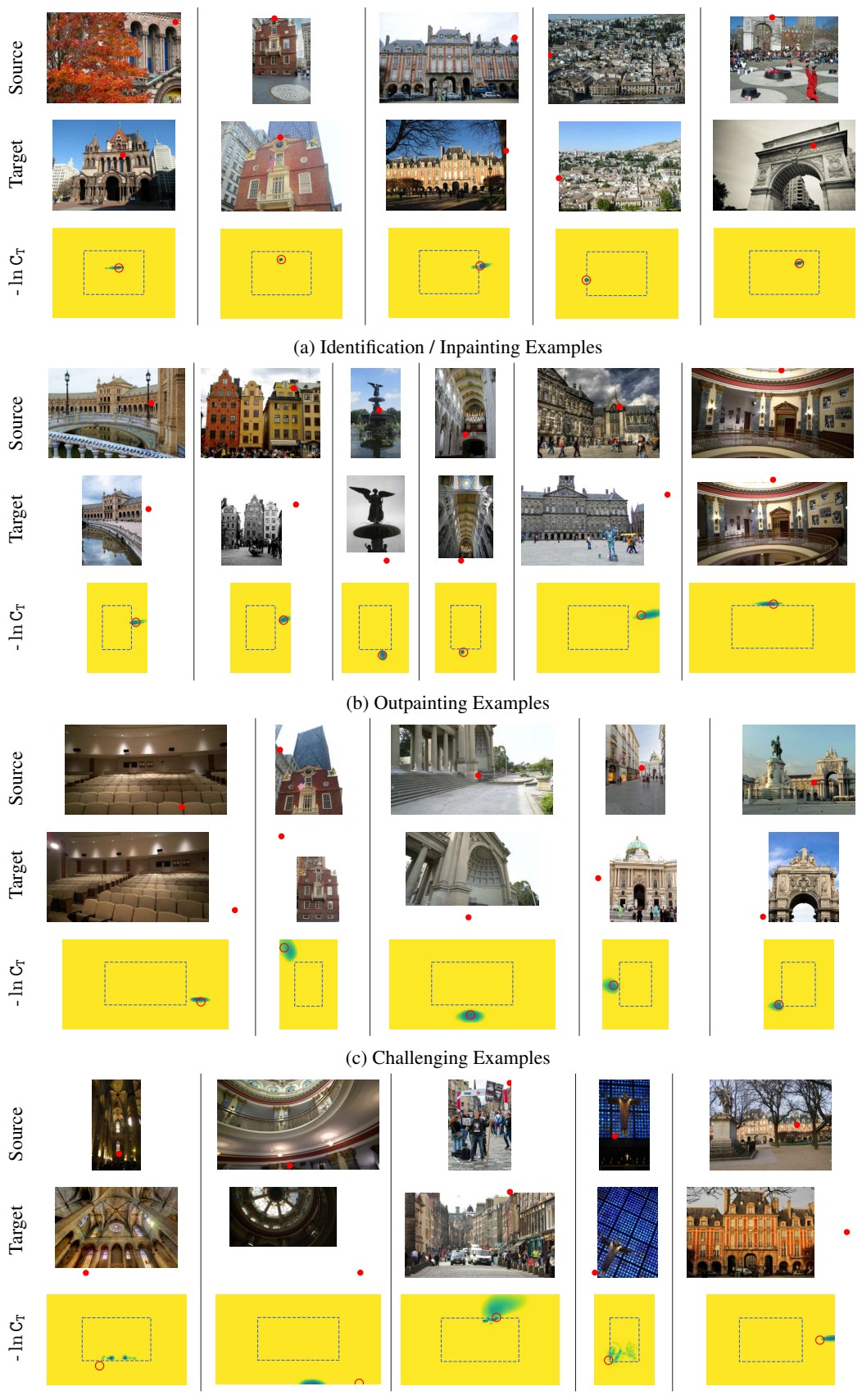

Figure 18: **Additional qualitative Megadepth (Li & Snavely, 2018) examples.** See text for details.

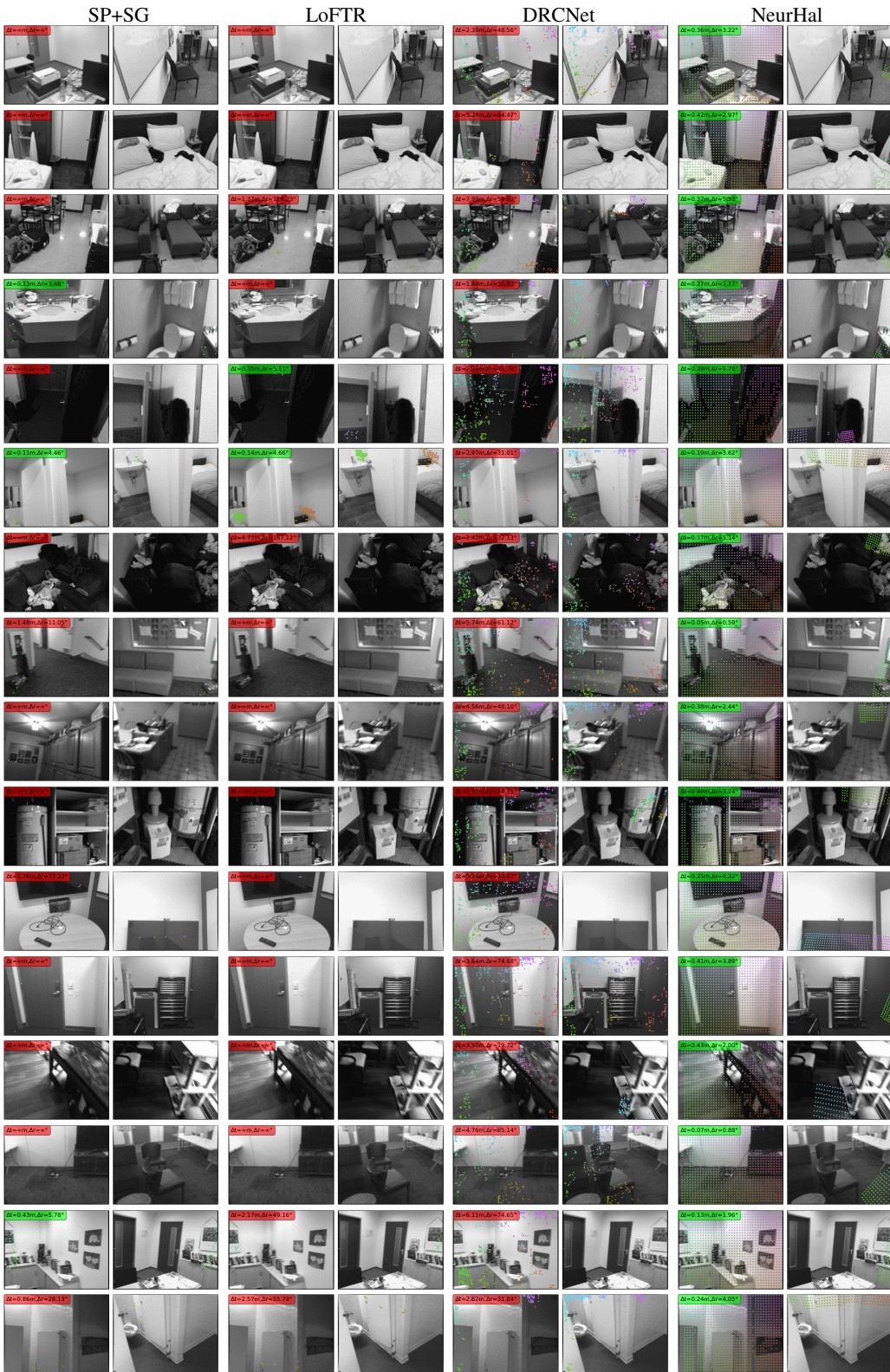

Figure 19: **Qualitative camera pose estimation results on low-overlap images from Scan-Net (Dai et al., 2017):** We show for every method keypoints used as input for the camera pose estimator in the source image (left image), along with their predicted reprojection in the target image (right image). We color-code keypoints based 2D spatial position in the source image. We also report for every pair and every method the camera pose estimation error in translation and rotation, colored in green when the pose is less than $\tau_t = 0.5m$ and $\tau_r = 10.0°$, and in red otherwise.

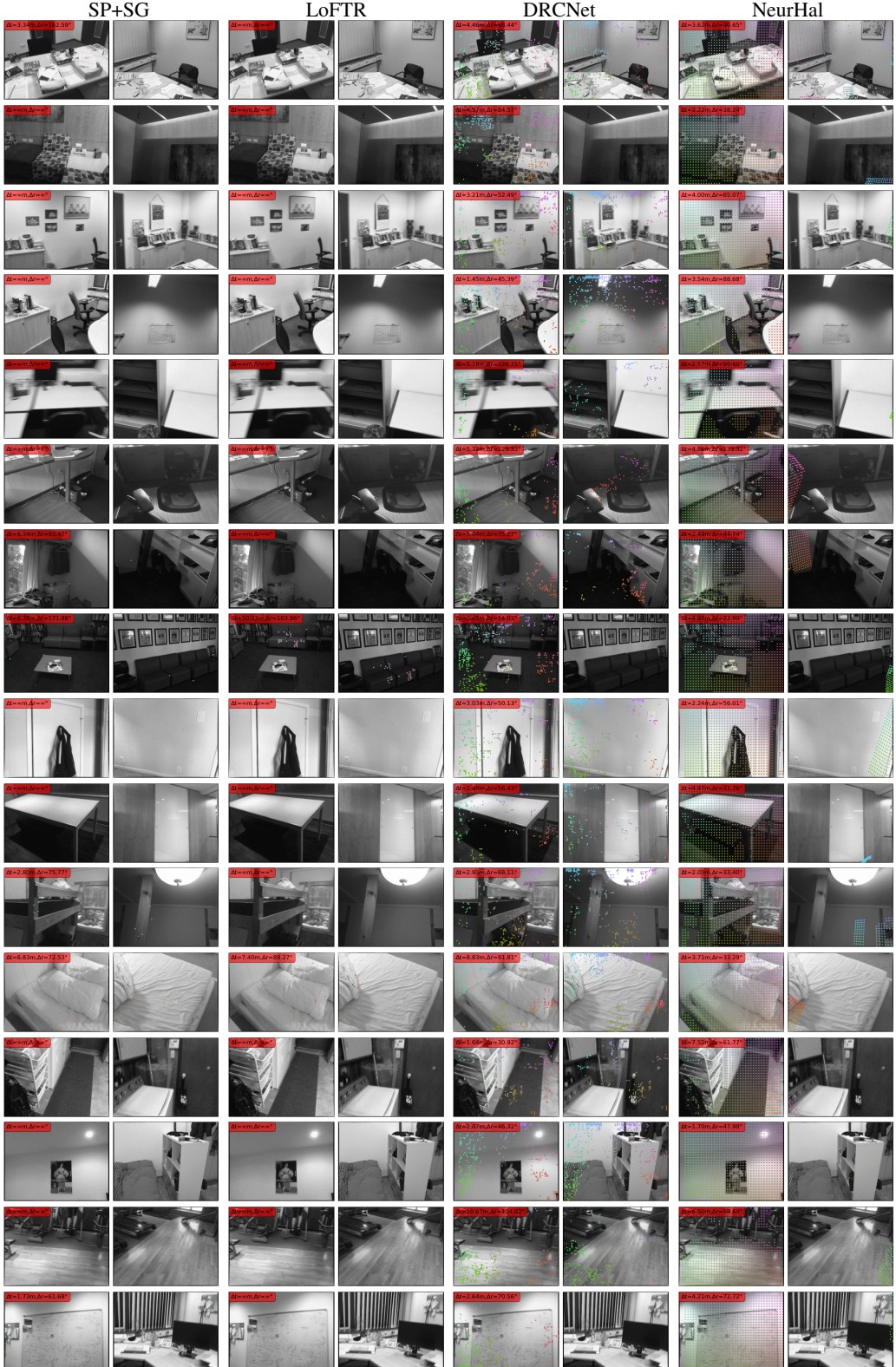

Figure 20: **NeurHal failure cases on low-overlap images from ScanNet (Dai et al., 2017):** We report cases where NeurHal fails to estimate a camera pose with an error less than $\tau_t = 0.5m$ and $\tau_r = 10.0°$. We find these cases often correlate with extremely low covisibility coupled with strong camera rotations.

