# OpenReview forum: "Visual Correspondence Hallucination"
_ICLR.cc/2022/Conference — ICLR 2022 Poster_

### Official Review · Reviewer_BNXs · 2021-10-24

**Correctness:** 3
**Technical Novelty And Significance:** 3
**Empirical Novelty And Significance:** 3
**Recommendation:** 8
**Confidence:** 3

**Main Review:**

Strengths:
(1) This work provides an interesting research direction correspondence learning, in which occluded and out of FoV correspondence can also help large-baseline camera pose estimation.
(2) Based on this insight: authors proposed a new model for hallucinating the correspondence based on attention blocks.
(3) Extensive experiments on indoor and outdoor benchmarks demonstrated that proposed model is able to hallucinate correspondence and outperform prior SoTA on wide baseline setting in terms of camera poes estimation.
(4) Author provided detailed supplementary material and trained model and source code for better supporting reproducibility.

Weakness:
(1) Authors admitted that the proposed method can only produce low resolution correspondence, which can have a negative impact on localization accuracy.
(2) Is there any explanation why correspondence inpainting is harder for inpainting?
(3) Did authors define how to compute Kc for transforming points from image plane to correspondence plane? It seems that I am not able to find it in the main manuscript.
(4) It would be better if authors could provide more analysis and ablation study on the relationship between accuracy of camera pose estimation and distribution of hallucinated correspondence. And in which case, hallucinated correspondence will be more accurate or less accurate.
(5) Authors should also cite several other related work for feature matching:
Learning feature descriptors using camera pose supervision, ECCV 2020
Self-Supervised Geometric Perception, CVPR 2021
Patch2Pix: Epipolar-Guided Pixel-Level Correspondences, CVPR 2021


**Summary Of The Paper:**

In this manuscript, authors proposed a new problem of correspondence hallucination, in which for keypoint in the source image, its correspondence should be detected regardless if it is occluded or outside the field of view. In particular, the authors proposed a new model and training paradigm that learn to hallucinate correspondence by predicting its probability distribution of its location. Extensive experiments on both indoor and outdoor benchmarks demonstrated that proposed method  can help camera pose estimation and outperforms prior state of the art feature matching approaches.


**Summary Of The Review:**

In summary, based on the strength and weakness I mentioned, I like this paper's idea for hallucinating correspondence so that we can obtain better accuracy for camera pose estimation, although it would be better to have more analysis and ablation study for the impact of correspondence hallucination on camera pose estimation accuracy.

---

> ### Author Response · Authors · 2021-11-11
> **Reply to BNXs**
>
> We would like to thank reviewer BNXs for his/her comments. Please find our answers below:
>
> - _(1) Authors admitted that the proposed method can only produce low resolution correspondence_: We would like to highlight that it is not the method itself that limits the resolution, but the implementation choices we had to make to train a network on 16GB memory GPUs in a reasonable amount of time (less than 24 hours in our case). By relaxing some of these constraints, there exists simple ways to significantly increase the resolution, such as gradient checkpointing (reduces the memory footprint but increases the training time) or using GPUs with larger memory.
>
> - _(2) Is there any explanation why correspondence inpainting is harder for inpainting?_: We believe the inpainted correspondence maps are less peaked than the outpainted correspondence maps because outpainting a correspondent essentially consists in transferring the features from location $p_{s,i}$ in the source features to location $p_{t,i}$ in the target features to obtain $d_{s,i} = D_{T,pad} (p_{t,i})$ (please refer to figure 2 to understand these notations). In the inpainting case, $p_{s,i}$ is occluded by an object in the target image, and this object is (often) also visible in the source image at location $p_{s,occ}$ . Thus, to produce peaked correspondence maps for both $p_s$ and $p_{s,occ}$ , the network has to output features such that $d_{s,i} = D_{T,pad} (p_{t,i}) = d_{s,occ} $ which is more difficult than just $d_{s,i} = D_{T,pad} (p_{t,i})$. We added this paragraph in the updated version of the Appendix (Section A.2).
>
> - _(3) Did authors define how to compute Kc for transforming points from image plane to correspondence plane? It seems that I am not able to find it in the main manuscript._: We left out that computational detail from the manuscript. $K_c$ simply corresponds to a rescaled and offsetted transformation of the original calibration matrix, found by changing the focal length and offsetting the principal point of the camera. We added the analytical formula to compute $K_c$ in the Appendix (Section C.1).
>
> - _(4) It would be better if authors could provide more analysis and ablation study on the relationship between accuracy of camera pose estimation and distribution of hallucinated correspondence. And in which case, hallucinated correspondence will be more accurate or less accurate._: We would like to highlight that we performed an analysis in that direction in section B.2 where we report the percentage of camera poses being correctly estimated for several values of $\gamma$, which demonstrates the benefits of outpainting with a large $\gamma$ for camera pose estimation. In Figure 6 we also showed that learning to inpaint does not bring any significant improvement. Outpainting improves the camera pose because outpainted correspondences are outside the field of view and thus complement the identified correspondences, and thus better constrain the camera pose estimate. On the contrary, inpainted correspondences are usually surrounded by identified correspondences, thus the information they provide is redundant and does not allow to better constrain the camera pose estimate. We added this analysis in the Appendix (Section B.3).
>
> - _(5) Authors should also cite several other related work for feature matching_: Thank you for pointing out these missing references, they have been added to the updated version of the paper.
>
> We took into account the suggestions from all three reviewers and updated our submission accordingly.
> We also uploaded a video demonstrating the generalization performances of NeurHal on both indoor and outdoor smartphone-captured videos in the supplementary material.

---

> > ### Comment · Reviewer_BNXs · 2021-11-18
> > **Reply to the feedback**
> >
> > I read all the reviews and feedback. In general, I am OK with most of the explanations and also the updated versions is better for illustration of concerns I pointed out.  I think the technical contribution is sufficient (The proposed method is able to predict the location of correspondences for improving camera localization quality)

---

### Official Review · Reviewer_nobf · 2021-10-31

**Correctness:** 3
**Technical Novelty And Significance:** 2
**Empirical Novelty And Significance:** 2
**Recommendation:** 5
**Confidence:** 4

**Main Review:**

Strengths:
+The paper proposed an new problem: if and how neural network can predict unseen scene (keypoints) from a source-target image pair. Human is able to heuristically guess the location of outpainting correspondences from the geometry. For example, the keypoint displacement is relative camera pose and scene depth. This paper uses a deep model to accomplish this goal.
+ The paper proposed a method that is based on correspondence map and neural reprojection error [Germain 2021] with sufficient details.
+ NeurHal is tested on public datasets for precision of correspondences and camera pose estimation, showing better result than baselines.

Weakness:
- The proposed method has limited technique contribution. For example, the original NRE [Germain 2021]  is able to predict outpainting correspondences (it gives an extra category for un-matched keypoints). As a result, this paper adapts [Germain 2021] to solve this problem.
- In figure 3, the baseline (uniform correspondence map) is too weak. In figure 4, I do not find the precision for identified keypoints. It is unfair for the other methods.
- In camera pose estimation (section 4.2), the description of the baseline (correspondent to the first (light blue) method in figure 6) is not clear to me. Because some methods highly rely on good correspondences but others are not. So, it is hard to tell if the proposed method is able to improve the state-of-the-art camera pose estimation methods.

**Summary Of The Paper:**

The paper proposed a deep method (NeurHal) to predict visible, occluded or out-of-view keypoint matching from source images to target images. In training, a correspondence map is obtained from ground truth of camera matrix/pose, and images. In testing, the model directly outputs three categories of matchings: identified, outputpainting and inpainting. As an application of NeurHal, the method is applied to camera pose estimation and tested on ScanNet and Megadepth. The experiments shows the method improves the estimation accuracy, particularly the outputpainting correspondences.

**Summary Of The Review:**

I tend to reject the paper because of the lack of contribution in technique and insufficient experiments. I would like to see
1. the technique differences, compared with [Germain 2021] .
2. improve the baselines, and compare with the state-of-the-art camera pose estimation methods.

---

> ### Author Response · Authors · 2021-11-11
> **Reply to nobf**
>
> We would like to thank reviewer nobf for his/her comments. Please find our answers below:
>
> - _The proposed method has limited technique contribution. For example, the original NRE [Germain 2021] is able to predict outpainting correspondences (it gives an extra category for un-matched keypoints). As a result, this paper adapts [Germain 2021] to solve this problem_: We would like to highlight that [Germain 2021] is **not** able to predict outpainting correspondences. They do consider an extra category for un-matched keypoints but the probability for that category is set to **zero**, ie. the network does not output any score for that category (it is stated in section 3 of [Germain 2021]: "By definition, $C_{Q,n} (p_n^Q = out) := 0.$"). Setting this probability to zero is due to the fact that [Germain 2021] considers a classical siamese CNN architecture that does not allow the features of both images to communicate. [Germain 2021] is what we called, in the introduction, a "pure pattern recognition approach". Moreover, even if [Germain 2021] were using a non-siamese architecture, their method would output a single score for the category "un-matched keypoint" which would allow the network to detect when the correspondent is not visible but would not be sufficient to outpaint the **location** of the correspondent.
>
> - _In figure 3, the baseline (uniform correspondence map) is too weak_:  Figure 3 illustrates the difference in NRE distributions for different kinds of hallucinated correspondences. In this plot the uniform distribution label is not a baseline, but merely an indicator which serves as a reference point to interpret the NRE values and argmax distances. Comparison against state-of-the-art methods (baselines) can be seen in Figure 4.
>
> - _In figure 4, I do not find the precision for identified keypoints. It is unfair for the other methods._: As stated in the section "contributions", our paper seeks to answer to the question: "Can we derive a network architecture able to learn to hallucinate correspondences?". Thus in Figure 4, we evaluated the performances on inpainting and outpainting to highlight that sota methods are not able to hallucinate correspondences while NeurHal can. We already know that all these methods are very good at identifying correspondences. In Figure 3, we did compute the identification histograms of NeurHal to highlight/compare the difficulty of inpainting and outpainting with respect to the task of identification.
>
> - _In camera pose estimation (section 4.2), the description of the baseline (correspondent to the first (light blue) method in figure 6) is not clear to me._: The baseline of Figure 6 corresponds to training our model using only identifiable (ie. covisible) keypoint correspondences. The other curves incorporate additional training data coming from non-covisible correspondences (inpainting and outpainting). We have made this clearer in the updated version of the paper.
>
> - _So, it is hard to tell if the proposed method is able to improve the state-of-the-art camera pose estimation methods._: Our experiments on absolute camera pose estimation demonstrate our method (trained jointly on identified, inpainted and outpainted correspondences) is indeed able to significantly improve the state-of-the-art, in the case of low-overlap image pairs (please see Figure 7, 12 and 13). Could you please specify why Figure 7, 12 and 13 do not clearly demonstrate the improvement of NeurHal over the state-of-the-art camera pose estimation methods?
>
> - _[...] improve the baselines, and compare with the state-of-the-art camera pose estimation methods._: To the best of our knowledge we did compare against the state-of-the-art camera pose estimation methods (LoFTR, DRCNet, SP+SG, R2D2, S2D). These are the best performing methods on visual localization benchmarks (e.g. [visuallocalization.net](https://www.visuallocalization.net/)).
>
> We took into account the suggestions from all three reviewers and updated our submission accordingly.
> We also uploaded a video demonstrating the generalization performances of NeurHal on both indoor and outdoor smartphone-captured videos in the supplementary material.

---

> > ### Comment · Reviewer_nobf · 2021-11-17
> > **Reply to the feedback**
> >
> > I read all reviews and feedback. Here is my new comments, correspondence to the 6 items above.
> > 1. It is a good point. Please highlight that NeurHal can output "location" of outpaint.
> > 2. It is Ok to say it is not a baseline. From geometric perspective, an non-learning baseline could be: 1. compute a homography from the source image to the target image using 4-point-method and RANSAC. 2. mapping all keypoints from source image to target image. Some of the keypoints will be out of the boundary of the target image. This baseline has a couple of assumptions: low depth variance of the scene, sufficient number keypoint to perform the homography estimation. However,  it is better than the uniform correspondence map.
> > 3. Figure 4 highlights the paper's contribution on inpainting and outpainting keypoints. Why not show the result for identified keypoints. In applications such as 3D reconstruction, the matching accuracy of identified keypoints are more important than non-identified keypoints which are excluded in the rest of the pipeline line.
> > 4. It is clear to me now.
> > 5 and 6: I checked the score table in the link. LoFTR, DRCNet, SP+SG, R2D2, S2D are indeed stat-of-the-art. However, still need to clarify 1. Do you perform the same task as in the https://www.visuallocalization.net/benchmark/? From the writing, the "absolute-pose" is actually the relative pose between the source image view and target image view.
> > 2. In Figure 7, you choose a different threshold (1.5m and 20^o) from these in the  score board  ((0.25m, 2°) / (0.5m, 5°) / (5m, 10°)). The angular error threshold is much bigger than the standard (~5^o). Why?
> >
> > I do not expected all concerns be addressed, but I would like to see if author can enhance the baseline, and clarify the experiment settings.

---

> > > ### Author Response · Authors · 2021-11-22
> > > **Reply to nobf**
> > >
> > > Thank you for your reply, please find our responses below:
> > >
> > > 1. _It is a good point. Please highlight that NeurHal can output "location" of outpaint._
> > >    We added a paragraph discussing our approach versus [Germain 2021] in Sec. A.5 of the updated version of the paper.
> > > 2. _It is Ok to say it is not a baseline. From geometric perspective, an non-learning baseline could be: 1. compute a homography from the source image to the target image using 4-point-method and RANSAC. 2. mapping all keypoints from source image to target image. Some of the keypoints will be out of the boundary of the target image. This baseline has a couple of assumptions: low depth variance of the scene, sufficient number keypoint to perform the homography estimation. However, it is better than the uniform correspondence map._
> > >    We added an experiment that uses homography estimates to warp source keypoints as you suggested in Sec. A.2 and Fig. 9 of the updated version of the paper.
> > > 3. _Figure 4 highlights the paper's contribution on inpainting and outpainting keypoints. Why not show the result for identified keypoints. In applications such as 3D reconstruction, the matching accuracy of identified keypoints are more important than non-identified keypoints which are excluded in the rest of the pipeline line._
> > >    We agree that 3D reconstruction is a very important application, however performing 3D reconstruction from coarse correspondence maps is not straightforward. In the paper we focused on hallucinating correspondences using a low-resolution correspondence map setting, which prevents NeurHal from predicting highly accurate correspondences. Thus we left as future work the relevance of applying NeurHal to 3D reconstruction.
> > > 4. _It is clear to me now. 5 and 6: I checked the score table in the link. LoFTR, DRCNet, SP+SG, R2D2, S2D are indeed stat-of-the-art. However, still need to clarify 1. Do you perform the same task as in the https://www.visuallocalization.net/benchmark/? From the writing, the "absolute-pose" is actually the relative pose between the source image view and target image view._
> > >    We would like to clarify we do follow the same task as the online benchmark: We perform absolute camera pose estimation (ie. we estimate the rotation matrix between the source and the target images as well as the full 3 D.o.F. translation vector between the optical centers of the source and target cameras). We call this task "absolute camera pose estimation" as the depth of the source image is known which allows us to estimate the full 3 D.o.F. vector. This is as opposed to the "relative camera pose estimation" problem where the depth is unknown, thus the scale of the translation is not observable and only the translation direction can be retrieved.
> > > 5. _In Figure 7, you choose a different threshold (1.5m and 20^o) from these in the score board ((0.25m, 2°) / (0.5m, 5°) / (5m, 10°)). The angular error threshold is much bigger than the standard (~5^o). Why?_
> > >    We would like to highlight that the choice of our thresholds are arbitrary, in fact the choice of thresholds in visual localization benchmarks is rarely backed by statistical evidence. In order to show that our thresholds are not too loose, we reported in Figure 13 the performance of NeurHal, state-of-the-art feature matching methods and the identity pose, on ScanNet for several rotation and translation thresholds. We can see that 1.5m / 20.0° is not an "easy" threshold because the identity pose obtains poor performances. We added this analysis in the updated version of the Appendix (C.3) and refered to it in the main paper.
> > >
> > > _I do not expected all concerns be addressed, but I would like to see if author can enhance the baseline, and clarify the experiment settings._
> > >
> > > We hope ours answers regarding the experiment settings as well as the homography baselines we added in the updated version of the paper will address your concerns.

---

### Official Review · Reviewer_uCMw · 2021-11-02

**Correctness:** 4
**Technical Novelty And Significance:** 4
**Empirical Novelty And Significance:** 3
**Recommendation:** 8
**Confidence:** 4

**Main Review:**

*Strengths*

The capability to predict the location of points that have just become occluded or exited the frame is useful in many tasks, such as perception for robots where such predictions could guide the robot’s next moves. Humans are good at this, but computer vision research has focused on the limited setting when the keypoints are visible in both images. Loss functions used for training networks for correspondence estimation only consider descriptors of visible keypoints. This is an important contribution of the paper and it goes beyond more qualitative work of predicting unseen objects based on context.

The approach builds upon the concept of Neural Reprojection Error (NRE) which was recently introduced by Germain et al. (2021). NRE does not require that the appearance of two image regions, that are hypothesized to be projections of the same 3D point, is similar. It can be viewed as an extension of the purely geometric reprojection error. Germain et al. did not consider occluded or out-of-bounds keypoints.

I consider the unified treatment of visible, occluded and out-of-bounds keypoints an advantage.

Experiments are thorough, findings are explained clearly, and limitations of the proposed method are pointed out. I consider the use of four publicly available datasets, two indoor and two outdoor, sufficient. Over one million keypoints are used in the experiments.

Generalization on additional datasets not used for training is also shown. This is important since it allows broader deployment of the algorithm.

The main limitation of NeurHal is low accuracy under favorable conditions, due to the low resolution of its output. (Improving this is left as a direction for future work.)

The appendices contain useful additional information, experiments and implementation details.

*Weaknesses*

It is unclear to me what the network actually learns. I speculate that it learns to warp the source image to the target given the relative pose between the two cameras and the depth of the keypoint in the source image. The output of this process is a correspondence map, which suggests that multiple warpings are considered. This is not described clearly enough. (The fact that the visibility of keypoint in the target images does not need to be labeled is clear.)

The metrics used for evaluation are somewhat arbitrary, or not sufficiently justified. Figure 3, for example, compares the results to random chance, which is a very weak baseline. Camera pose estimation is considered correct when the rotation error is under 20 degrees and the translation error is under 1.5 m. The latter is uninformative without knowing the magnitude of the translation between the two cameras. 1.5 m may be a very small or a very large error depending on the input data.


*Other Comments*

I find the term hallucination in the title marginally acceptable, but I think that it is abused in the rest of the paper. For example in: “Local feature matching methods are only able to identify the correspondent’s location when it is visible, while humans can also hallucinate its location when it is occluded or outside the field of view through geometric reasoning.” Humans do not hallucinate, they predict or estimate.

Footnote 1 on p. 3 is unnecessary. The equation it refers to holds for any images in general configuration sharing the same intrinsic parameters (K matrix). The latter constraint can be easily relaxed.

Section 3.2: the calibration matrix K_c does not encode the boundaries of the image. Instead of a different matrix, what is needed are different boundary conditions for considering pixels to be within the image or not.

Section A.2 that discusses additional related work should be moved to the main paper. It is very relevant to the problem at hand.

Figure 8: precision on the y-axis is perplexing. The description suggests that this should be recall.



**Summary Of The Paper:**

The paper proposes a method that accepts two partially overlapping images and a keypoint in one of them, and detects the location of the keypoint in the other image, regardless of whether it is visible, occluded, or outside the frame of the image. This is an interesting re-formulation of the correspondence estimation problem, the conventional formulation of which considers only keypoints that are visible in both images. Under the previous formulation, correspondence estimators are considered successful if they can declare that the correspondent of the keypoint is invisible in the target image. In addition to the capability to predict the location of invisible keypoints in isolation, the paper demonstrates that this capability is beneficial to camera pose estimation.

**Summary Of The Review:**

The paper contains an important contribution as discussed above. The fact that predicting the locations of invisible points improves camera pose estimation provides further support that the method is useful in downstream tasks.

---

> ### Author Response · Authors · 2021-11-11
> **Reply to uCMw**
>
> We would like to thank reviewer uCMw for his/her comments. Please find our answers below:
>
> - _It is unclear to me what the network actually learns. I speculate that it learns to warp the source image to the target given the relative pose between the two cameras and the depth of the keypoint in the source image. The output of this process is a correspondence map, which suggests that multiple warpings are considered. This is not described clearly enough._: We agree with this analysis. However as you correctly pointed out, this is a "speculation" as there is no proof the network is actually doing this, so at the time of submission we decided not write anything about it. As requested, we updated our submission and added a paragraph in the Appendix (Sec A.2).
>
> - _The metrics used for evaluation are somewhat arbitrary, or not sufficiently justified. [...]  Camera pose estimation is considered correct when the rotation error is under 20 degrees and the translation error is under 1.5 m. The latter is uninformative without knowing the magnitude of the translation between the two cameras_: We agree that the choice of our thresholds are arbitrary, in fact the choice of thresholds in visual localization benchmarks is rarely backed by statistical evidence. In order to show that our thresholds are not too loose,  we reported in Figure 13 the performance of NeurHal, state-of-the-art feature matching methods and the identity pose, on ScanNet for several rotation and translation thresholds. We can see that 1.5m is not an "easy" threshold because the identity pose obtains poor performances. We added this analysis in the updated version of the Appendix (C.3) and refered to it in the main paper.
>
> - _Figure 3, for example, compares the results to random chance, which is a very weak baseline_: In this plot the uniform distribution label is not a baseline, but merely an indicator which serves as a reference point to interpret the NRE values and argmax distances. Comparison against state-of-the-art methods (baselines) can be seen in Figure 4.
>
> - _I find the term hallucination in the title marginally acceptable [...] Humans do not hallucinate, they predict or estimate._: We employed the term "hallucinating correspondences" as it is a significantly different task from "identifying correspondences". We believe the terms "predicting correspondences" or "estimating correspondences" embed the identification and hallucination tasks. As requested we either removed some instances of the term "hallucination" in the text, or appended the word "predict" in the updated version of the paper.
>
> - _Footnote 1 on p. 3 is unnecessary_: We replaced $K$ by $K_S$ and $K_T$ in Sec. 3.2 and removed the footnote.
>
> - _Section 3.2: the calibration matrix K_c does not encode the boundaries of the image_: We agree the term "calibration matrix" is misleading, we changed it to "affine transformation matrix".
>
> - _Section A.2 that discusses additional related work should be moved to the main paper_: We moved the additional related work from the Appendix to the main paper in its updated version.
>
> - _Figure 8: precision on the y-axis is perplexing. The description suggests that this should be recall._: We removed the term precision from the paper.
>
> We took into account the suggestions from all three reviewers and updated our submission accordingly.
> We also uploaded a video demonstrating the generalization performances of NeurHal on both indoor and outdoor smartphone-captured videos in the supplementary material.

---

> > ### Comment · Reviewer_uCMw · 2021-11-13
> > **Response to author rebuttal**
> >
> > I have read all reviews and authors responses. I also looked at the additions to the paper mentioned here. The authors and I seem to be on the same page. The revisions to the paper are minor, but in the right direction.
> >
> > I will not insist on the use of "hallucination" since I am unable to propose a better alternative. I did not list it as a weakness.
> >
> > I find the discussion on what the network learns and on evaluation metrics acceptable. The former may be hard to decipher, and the metrics are presented clearly for readers to draw their own conclusions.
> >
> > Nitpicking: an affine transformation matrix does not encode image boundaries either. This may be a point where we are not on the same page.

---

> > > ### Author Response · Authors · 2021-11-16
> > > **Reply to uCMw**
> > >
> > > Thank you for your reply.
> > > Regarding the affine matrix $K_c$, we believe we misunderstood your initial comment.
> > > You are correct to point out $K_c$ does not encode the boundary conditions determining if a correspondent falls within the correspondence map or not. In practice these boundary conditions are computed using $\gamma$, initial image resolution and the image-to-correspondence map downscaling ratio. We explain how they are computed in section C.1. Please let us know if this update is satisfactory.

---

> > > > ### Comment · Reviewer_uCMw · 2021-11-18
> > > > **boundaries**
> > > >
> > > > Yes, it is satisfactory. This is a minor issue.

---

### Decision · Program_Chairs · 2022-01-20

**Decision:**

Accept (Poster)

**Comment:**

This paper receives positive reviews. The authors provide additional results and justifications during the rebuttal phase. All reviewers find this paper interesting and the contributions are sufficient for this conference. The area chair agrees with the reviewers and recommends it be accepted for presentation.